# TuneTables: Context Optimization for Scalable Prior-Data Fitted Networks

**Benjamin Feuer**[1], **Robin Tibor Schirrmeister**[2], **Valeriia Cherepanova** [*3], **Chinmay Hegde**[1],
**Frank Hutter**[2], **Micah Goldblum**[1], **Niv Cohen**[†1], **Colin White**[†4]

[1] New York University, [2] University of Freiburg, [3] University of Maryland, [4] Abacus.AI

## Abstract

While tabular classification has traditionally relied on from-scratch training, a recent breakthrough called prior-data fitted networks (PFNs) challenges this approach. Similar to large language models, PFNs make use of pretraining and in-context learning to achieve strong performance on new tasks in a single forward pass. However, current PFNs have limitations that prohibit their widespread adoption. Notably, TabPFN achieves very strong performance on small tabular datasets but is not designed to make predictions for datasets of size larger than 1000. In this work, we overcome these limitations and substantially improve the performance of PFNs via context optimization. We introduce TuneTables, a parameter-efficient fine-tuning strategy for PFNs that compresses large datasets into a smaller learned context. We conduct extensive experiments on nineteen algorithms over 98 datasets and find that TuneTables achieves the best performance on average, outperforming boosted trees such as CatBoost, while optimizing fewer than 5% of TabPFN's parameters. Furthermore, we show that TuneTables can be used as an interpretability tool and can even be used to mitigate biases by optimizing a fairness objective. We open-source our code and raw results at https://github.com/penfever/TuneTables.

## 1 Introduction

Tabular data, or data organized into rows and columns consisting of distinct features, are the oldest and one of the most ubiquitous types of data in machine learning in practice [10, 67]. Tabular data has numerous applications across medicine [41, 71], online advertising [33, 52, 64], finance [7, 18], and other areas [11, 12, 72].

Competitive classification algorithms for tabular data include gradient-boosted decision trees [16, 61] and deep neural networks [31, 42, 69]. Both approaches fit their respective models on a labeled dataset containing samples from a distribution reflecting the task at hand. A recent breakthrough, prior-data fitted networks (PFNs) [35, 55], are are a specific type of neural process which learn to perform approximate Bayesian inference in a single forward pass using in-context learning [50]. PFNs do not require optimizing parameters or fitting a model on downstream training data, instead feeding training data into the context and conditioning on it. In particular, TabPFN achieved state-of-the-art classification on small tabular datasets [35, 51].

The in-context learning approach of PFNs parallels that of large language models (LLMs) [82]. Both approaches can be viewed as approximate Bayesian inference, whether implicitly [79] or explicitly [55]. While researchers have successfully used various context optimization strategies for enhancing LLM performance [49], no prior work has studied context optimization strategies for PFNs. Furthermore, although TabPFN achieves very strong performance on small datasets, its limitations

---

[†]Equal advising. Correspondence to: bf996@nyu.edu.

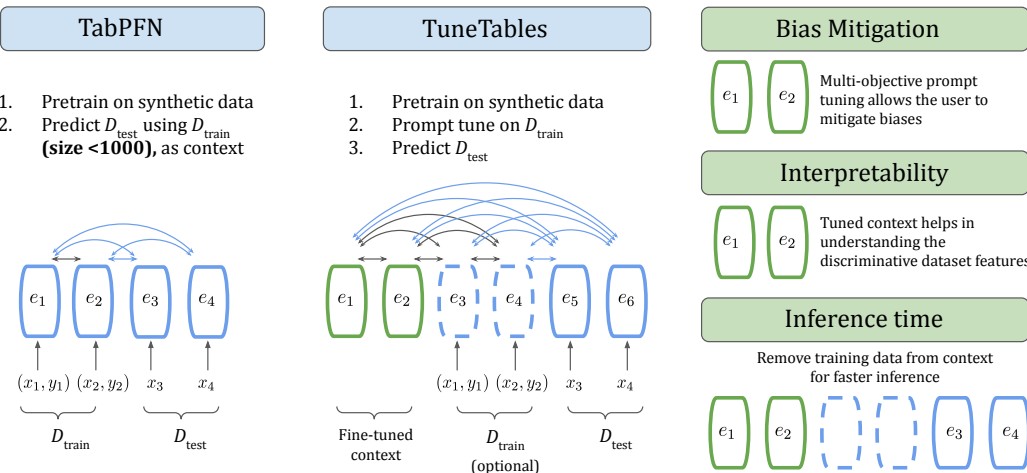

Figure 1: **TuneTables: a novel prompt-tuning technique for prior-data fitted networks.** TuneTables performs prompt tuning on a pre-trained prior-fitted network (TabPFN) to distill real-world datasets into learned embeddings, allowing for stronger performance and faster inference time than TabPFN in many cases. TuneTables also expands the capabilities of pre-trained PFNs; by way of example, we demonstrate its effectiveness for bias mitigation, and as an interpretability tool.

currently prohibit its widespread adoption: it only runs on datasets whose number of training samples, number of features, and number of classes are at most 1000, 100, and 10, respectively.

In this work, we perform the first investigation into context optimization strategies for PFNs, allowing us to substantially improve their performance when scaled to large datasets. Specifically, we introduce **TuneTables**, a novel parameter-efficient fine-tuning technique for PFNs that compresses large datasets into a smaller learned context (Figure 1). We conduct an extensive empirical investigation into TuneTables' performance on the TabZilla Benchmark Suite, the largest benchmark considered by recent tabular data literature [51]. Over 98 datasets and 19 algorithms, we find that TuneTables achieves the best average performance and is the best-performing method on 30 of them.

Because TuneTables effectively compresses the contents of large tabular datasets into the tuned prompt, no training data is needed in the context during inference, significantly speeding up inference time (similar to works on neural processes; see Section 6). We also show that the learned prompt can be used as a tool for interpretability. Finally, we show how to use TuneTables for multi-objective optimization, such as optimizing both accuracy and fairness simultaneously, allowing users to mitigate biased predictions of a pretrained PFNs with just a lightweight tuning procedure. We open-source our code and raw results at https://github.com/penfever/TuneTables.

**Our contributions.** We describe our main contributions below.

- We introduce **TuneTables**, a parameter-efficient fine-tuning technique for PFNs. TuneTables achieves the highest number of wins (30) when compared to 19 algorithms over 98 datasets, while requiring less inference time than TabPFN.
- We show how to use prompt tuning for **multi-objective optimization**, such as optimizing both accuracy and fairness simultaneously, allowing users to mitigate biased predictions of a pretrained PFNs with just a lightweight tuning procedure.
- We show that TuneTables' condensed representations can be used as an **interpretability** tool.
- We conduct an extensive study on context optimization strategies for PFNs by performing an ablation study on TuneTables, as well as studying sketching and feature selection techniques.
- In order to better manage the limitations of our method, we introduce TuneTables-medium and TuneTables-light, which achieve strong tradeoffs between precision and speed.

## 2   Background

**PFNs: review and limitations.**   In this section, we give a background on PFNs and discuss their limitations. For a complete description, see [35, 55, 58]. Assume that we have a classification problem with features $\mathcal{X} \subseteq \mathbb{R}^d$ and labels $\mathcal{Y}$. Given a dataset $D = D_{\text{train}} \cup D_{\text{test}}$, where $D_{\text{train}} = \{(x_1, y_1), \ldots, (x_n, y_n)\}$ and $D_{\text{test}} = \{(x_{\text{test}}, y_{\text{test}})\}$, our goal is to predict the conditional class probabilities $p(\cdot \mid x_{\text{test}})$. In the Bayesian framework for supervised learning, the mechanism for generating the data distribution is a hypothesis $\varphi$, drawn from $\Phi$, the space of all hypotheses. $\Phi$ encodes our prior beliefs on the system before observing data. In this framework, datasets $D$ are generated by first drawing $\varphi \sim \Phi$, and then drawing *i.i.d.* samples according to $\varphi$. The posterior predictive distribution (PPD) for a test sample $x_{\text{test}}$ is the label distribution $p(\cdot \mid x_{\text{test}}, D_{\text{train}})$ that follows from our prior. We can obtain the PPD by integrating over the space of hypotheses $\Phi$:

$$p(y \mid x, D) \propto \int_\Phi p(y \mid x, \varphi) p(D \mid \varphi) p(\varphi) d\varphi. \tag{1}$$

A PFN is a transformer-based architecture trained to approximate the PPD via *synthetic prior-fitting*. Given a prior, we first sample hypotheses $\varphi \sim p(\varphi)$ and then synthetic datasets $D \sim p(D \mid \varphi)$. We optimize the parameters of the PFN by predicting the class labels of $D_{\text{test}} \subseteq D$, conditioned on $D_{\text{train}} = D \setminus D_{\text{test}}$. We compute the loss by:

$$\mathcal{L}_{\text{PFN}} = \mathbb{E}_{D \sim p(D)} \left[ -\log q_\theta(y_{\text{test}} \mid x_{\text{test}}, D_{\text{train}}) \right], \tag{2}$$

for simplicity assuming all training and test sets are size $n$ and 1, respectively. We then approximately solve this optimization problem, $\hat{\theta} = \arg\min_\theta \mathcal{L}_{\text{PFN}}$, allowing $q_\theta$ to approximate Equation (1):

$$q_\theta(y_{\text{test}} \mid x_{\text{test}}, D_{\text{train}}) \approx p(y_{\text{test}} \mid x_{\text{test}}, D_{\text{train}}). \tag{3}$$

**Scaling challenges.** While PFNs, specifically TabPFN, have shown remarkable success in classification by in-context learning, several important obstacles constrain their more widespread adoption:

1. **PFNs only accept a fixed number of features.** The current design of PFNs fixes the quantity of features at the time of pretraining. This quantity cannot be changed without retraining the PFN.
2. **PFNs scale poorly with the dataset size**. While PFN accuracy can improve with more real-data samples at inference time [35, 58], the memory requirements scale with the context length, making extensions beyond a certain number of samples impractical.
3. **PFNs only select from a fixed number of classes**. The MLP decoder head co-trained with the PFN fixes the number of classes that can be identified at test time.

**A motivating example.**   In Figure 3 (left), we present a comparison between CatBoost and TabPFNs3000 from [51], a version of TabPFN that, for datasets above 3000 data points, uses a random subset of 3000 data points, for fitting; this version also evaluates 30 feature subsets based on mutual information (requiring $30\times$ more time). We ablate our feature and sample subselection strategies for TabPFNs3000 in Appendix Table 13. Although TabPFNs3000 performs very well on datasets with fewer than 1000 datapoints and 100 features, it significantly underperforms CatBoost beyond those constraints.

**Approach.**   We propose sketching, feature selection, and fine-tuning as an attempt to remedy *(1)* and *(2)*. Then in Section 3, we describe novel prompt-tuning and class-extension strategies to create TuneTables, a robust classifier which remedies *(1)-(3)*.

### 2.1   Classical sketching and feature selection

**Sketching.**   The number of samples that a PFN can handle is limited to around 3000 by conventional GPU sizes. However, in the real world, datasets are often much larger. Going forward, we refer the maximum allowable context size of the PFN as $n$, and to the size of the real-world dataset as $N$.

Given a training dataset $D_{\text{train}}$ of size $N >> n$, one option is to select a representative subset of the dataset, $D_{\text{compact}} \subseteq D_{\text{train}}$, to use as the context. In general, researchers have studied a variety of data

summarization techniques, often called *sketching*, for tabular data [56]. In the context of a pretrained PFN $q_\theta$, the goal is to find a sketching function $s : \mathbb{R}^{N \times d} \mapsto \mathbb{R}^{n \times d}$ such that

$$
\mathbb{E}_{D \sim p(D)} \left[ -\log q_\theta(y_{\text{test}} \mid x_{\text{test}}, s(D_{\text{train}})) \right] \tag{4}
$$
$$
\approx \mathbb{E}_{D \sim p(D)} \left[ -\log q_\theta(y_{\text{test}} \mid x_{\text{test}}, D_{\text{train}}) \right].
$$

Here, we consider three sketching methods for TabPFN: *random*, in which we select a random subset of $n$ datapoints from the full training set; *k-means*, in which we compute the $n$-means clustering [3] of $D_{\text{train}}$ and select the $n$ centers; and *CoreSet*, in which we compute a core-set of size $n$ [2].

**Feature selection.**    In addition to the limitation on dataset size, PFNs, such as TabPFN, also impose a limitation on the number of features $d$. Similar to sketching, given a dataset with $D >> d$ features, we can perform feature selection or summarization in order to adhere to the constraint (formally, finding a function that reduces the feature dimension and approximates an expression similar to Equation (4)). Feature selection is a critical part of tabular classification, and there are several popular feature selection methods [13, 17]. We investigate three different methods: *random*, in which we randomly select a set of $d$ features; *mutual information*, in which we select $d$ features with the highest mutual information of the target dataset [75]; and *principal component analysis (PCA)*, in which we take the $d$ first principal components. In Section 4, we find that all sketching and feature selection methods plateau in performance well before approaching parity with GBDTs, which motivates our investigation of new scaling techniques.

## 2.2  Fine-tuning

We conclude this section with a discussion of another potential approach for scaling PFNs: **fine-tuning**. In this approach, given a dataset of size $N >> n$, we use gradient descent to continue training *all parameters* of the PFN. However, we show in Section 4 that fine-tuning takes up considerable memory resources while still not achieving competitive accuracy on large datasets; we postulate that for most datasets, the synthetic prior of TabPFN is more robust to overfitting than the actual training data; fine-tuning overfits to the validation set. This is borne out by the observation that when fine-tuning TabPFN, validation accuracy continues to improve even as test accuracy declines. To remedy this, we present new parameter-efficient solutions for scaling PFNs in the next section.

## 3  TuneTables

Motivated by the strong performance of prompt tuning, we introduce **TuneTables**, a new tabular classification algorithm. Using a pretrained PFN (TabPFN, in our experiments) as a base model, TuneTables overcomes the limitations of PFNs, allowing them to be applied to any tabular classification problem.

For the full details of the algorithm, please refer to Appendix D. We summarize here: *(a)* if a dataset is small enough to run with the original zero-shot version of TabPFN, we include it in our search, as the TabPFN is already highly optimized for such datasets; The reason why we do not only use TabPFN is that the exact size at which TuneTables outperforms TabPFN is dataset-dependent (with an average transition of around 800 samples). *(b)* if there are more than 100 features, we perform grid search over a set of feature subselection methods and select the one which performs best on the validation set; *(c)* orthogonally, if there are too many labels, we fit a new decoder to a frozen TabPFN; *(d)* we optimize over a search space of tuned prompts, both with and without real-data context during training, fitted to the dataset; *(e)* we report the best-performing model according to accuracy.

**Prompt tuning as a scalable context for PFNs.**    Motivated by the limitations of sketching for large contexts, we explore *soft prompt tuning* as an alternative. In soft prompt tuning [46], given a dataset $D = D_{\text{train}} \cup D_{\text{test}}$, a parameterized matrix $D_{\text{tune}}^{p \times e}$ is prepended to the input embedding $D_{\text{train}}^{n \times e}$, where $e$ is the transformer embedding dimension and $p$ is a hyperparameter–the size of the tuned prompt.

The paper [46] demonstrates the effectiveness of soft prompt tuning for NLP tasks by prepending a small task-specific prompt ($p \approx 5$). These task-specific learned tokens prove effective at the extremely large model scales commonly encountered in NLP. Somewhat surprisingly, we show that prompt tuning is effective at similar scales of $p$ even for the much smaller tabular data models (see Appendix D and Appendix F). However, prompt tuning increases in effectiveness when $p$ is larger.

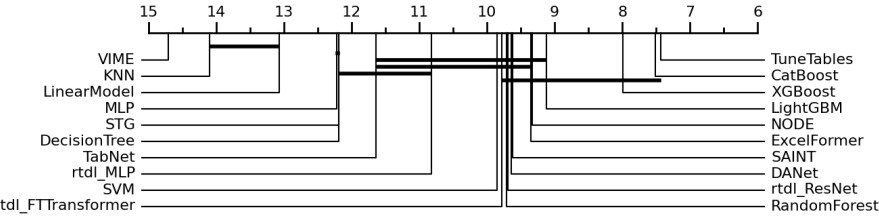

Figure 2: **TuneTables and state-of-the-art tabular models.** A critical difference plot according to mean accuracy rank across the 98 datasets in Table 1 of [51]. Algorithms which are *not significantly different* ($p > 0.05$) are connected with a horizontal black bar. TuneTables achieves the highest mean rank of any algorithm.

**Soft prompt tuning for tabular data.** Unlike in NLP, transformers for tabular data, including PFNs, generally accept two input embeddings; a continuous-valued embedding $D_{\text{train } X}$, and a categorically-valued $D_{\text{train } y}$ which is passed through directly. We adapt the method of [46] by fitting the parameters of $D_{\text{tune } X}^{p \times e}$ to $D_{\text{train } X}$ and randomly initializing $D_{\text{tune } y}^{p \times 1}$ with an equal number of labels from each class in $D_{\text{train}}$. These synthetic datapoints are optimized on the entire labeled set, and therefore give PFNs access to a much larger training set not accessible by existing methods.

We further adjust the method of [46] to allow for the possibility that $D_{\text{tune}}$ has learned, in essence, a distilled version of $D_{\text{train}}$; at test time, we evaluate two settings, hereafter referred to as $C$ ('context') and $NC$ ('no context'). In the $C$ setting, following [46], we have $D_{\text{tune}}^{p \times e}$ prepended to the input embedding $D_{\text{train}}^{n \times e}$. In the $NC$ setting, we provide only $D_{\text{tune}}^{p \times e}$ at test time. In Section 4, we empirically evaluate both approaches. We ablate specifics of our implementation choices in Appendix D. Unlike prompt tuning for NLP, we show that the $NC$ setting is often competitive with, if not better than, the $C$ setting. We also ablate this choice during training; see Appendix D for implementation specifics.

**Extending the number of predicted classes.** TabPFN uses a pretrained transformer, with a two-layer MLP as a final classifier. The pretraining procedures limit the naïve use of TabPFN to classification tasks with at most 10 classes. Yet, in many cases, datasets of interest might contain a larger number of classes, which would require pretraining a new PFN from scratch.

Following the method of last-layer retraining [44, 45], for such datasets, we fit a PFN to new posteriors given by real-world classification datasets with more than 10 classes, freezing all weights except for those of the decoder MLP and the tuned prompt (see Appendix D for more implementation details). Even after this modification, our method remains highly parameter-efficient, optimizing fewer than 5% of TabPFN's parameters.

## 4 Experiments

**Algorithms and datasets used.** We compare TuneTables to nineteen algorithms, including three GBDTs: CatBoost [61], LightGBM [43], and XGBoost [16]; 11 neural networks: DANet [14], FT-Transformer [31], two MLPs [31], NODE [60], ResNet [31], SAINT [69], STG [80], TabNet [6], TabPFN [35], VIME [81], and ExcelFormer [15]; and five baselines: Decision Tree [62], KNN [21], Logistic Regression [22], Random Forest [48], and SVM [20]. For all algorithms, we use their official implementation; see Appendix C for more details. We also compare to TabPFNs3000. We run the algorithms on the TabZilla Benchmark Suite introduced in [51]. This suite consists of 98 classification datasets from OpenML [74] with a diversity of sizes and number of features [51]. See Table 4 in Appendix C for a list of all datasets with their statistics.

**Experimental setup.** For all algorithms other than TuneTables, we perform light hyperparameter tuning by running one default setting and 29 iterations of random search using Optuna [4]); see Appendix C for details. Following [51], all of the algorithms come with their default set of hyperparameters used in the official implementation, and we used all of these settings. For TuneTables, we optimize via a grid search described in Appendix D.

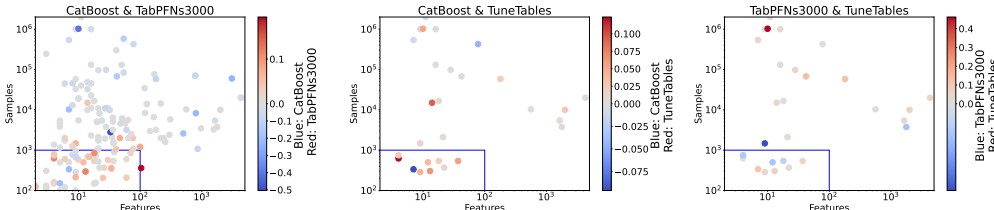

Figure 3: **TuneTables addresses TabPFN's limitations.** (Left) Motivating example (using the subset of [51]on which both CatBoost and TabPFNs3000 report results): TabPFNs3000 is best on small datasets, but when scaled past 3000 datapoints and 100 features, TabPFNs3000 significantly underperforms. (Middle) CatBoost vs. TuneTables on LARGESCALETABLES : By contrast, TuneTables is competitive with CatBoost on all datasets, mitigating the limitations of TabPFN. (Right) TabPFNs3000 vs. TuneTables on LARGESCALETABLES : TuneTables outperforms TabPFNs3000 on datasets with a high number of datapoints or features. The colorbar on the y axis represents the comparative change in per-dataset accuracy between two algorithms (A: blue, B: red). Positive numbers represent the absolute gain in accuracy of B w.r.t. A, negative numbers represent the absolute gain in accuracy of A w.r.t. B.

We fit for up to 100 epochs with early stopping. For all algorithms, we report the test performance of the hyperparameter set with the best performance on the validation set and cross-validate on three train/test folds from OpenML. We conduct our experiments on an NVIDIA L4 TPU with 24GB VRAM. We summarize our results across datasets by reporting mean accuracy as well as the mean normalized accuracy (after Z-score normalization) for each algorithm.

**Algorithm comparison.** We compute *statistically significant* performance differences among algorithms averaged across all datasets, as done in prior work [42]. We first use a Friedman test to check whether performance differences between all algorithms are significant [30]. We reject the null hypothesis for $p < 0.05$. Then we use a Wilcoxon signed-rank test to check which pairs of algorithms have significant performance differences [19]. We use a Holm-Bonferroni correction to account for multiple comparisons [36]. See Figure 2. We find that TuneTables achieves the highest average rank, although the difference among the top three algorithms is not statistically significant. In Table 1, we present the accuracy, rank, and Z-score of all algorithms, averaged across all datasets, and find that TuneTables performs very well on all metrics but runtime.

**Large datasets.** One limitation of the TabZilla Benchmark Suite is that even the largest dataset in the comparison contains only 45,211 samples. This scale is modest by the standards of modern tabular problems. In order to better understand the performance of TuneTables on extremely large datasets, we curate from [51] a novel benchmark, LARGESCALETABLES , consisting of 29 datasets with up to 1 900 000 samples, and 7200 features. We curate the datasets from OpenML, omitting image classification datasets. Since TabPFN generally outperforms boosted trees on these smaller datasets, and TuneTables extends TabPFN, we also heuristically select smaller datasets to LARGESCALETABLES so as not to favor either TabPFN or boosted trees. TuneTables achieves the highest average accuracy of any algorithm, and achieves the best performance on `poker-hand`, a dataset of size 1 025 009. The complete results can be found in Appendix Table 8 and Figure 3 (right). In order to assess the performance of TuneTables on datasets with many classes, we curate another subset of [51], presenting results on fifteen datasets with up to 100 classes. In appendix Table 7 we show that despite the large divergence from the PFN's pretraining, TuneTables achieves a mean rank of 2.52, ahead of CatBoost and a ResNet, second only to XGBoost, whose mean rank is 2.0.

**Runtime comparison.** We divide our consideration of runtime into inference and training. At inference time, TuneTables is around 9x faster than TabPFNs3000, on average; see Appendix Table 10 for the per-dataset details. However, the end-to-end runtime of TuneTables is over 7x that of CatBoost and XGBoost, and also slower than TabPFNs3000, because of the increased *training time*.

In order to better understand the trade-off between accuracy and runtime, we introduce efficient variants of our method. *TuneTables-medium* utilizes a more efficient adaptive sequence size (i.e., the number of real data points received at train time) which scales with the size of the dataset, validates

Table 1: **TuneTables matches SOTA algorithms on 98 datasets.** In this table, we compare algorithms over the 98 datasets in the TabZilla benchmark suite from [51]. For each algorithm, we compute its mean accuracy and mean rank in terms of accuracy. We also compute the mean Z-score, computed by normalizing the set of results on each dataset (by mean 0 std. 1), so that each dataset has the same weight, and averaging each algorithm's normalized performances. Std. Z-Score is computed with respect to random splits and averaged across datasets. Num. wins values are averaged over three splits per dataset.

| Model | Mean Acc. | Mean Rank | Mean Z-Scores | Std Z-Scores | Med Z-Scores | Num. Wins |
|---|---|---|---|---|---|---|
| TuneTables | **0.860** | **7.435** | 0.494 | **0.624** | 0.490 | **30** |
| CatBoost | 0.856 | 7.514 | **0.496** | 0.669 | **0.566** | 13 |
| XGBoost | 0.854 | 7.991 | 0.411 | 0.783 | 0.533 | 16 |
| ExcelFormer | 0.847 | 9.349 | 0.212 | 0.863 | 0.384 | 7 |
| LightGBM | 0.845 | 9.122 | 0.284 | 0.894 | 0.431 | 20 |
| RandomForest | 0.843 | 9.713 | 0.206 | 0.776 | 0.374 | 4 |
| SAINT | 0.840 | 9.619 | 0.132 | 0.985 | 0.337 | 10 |
| DANet | 0.840 | 9.646 | 0.209 | 0.763 | 0.345 | 3 |
| rtdl_ResNet | 0.839 | 9.691 | 0.175 | 0.798 | 0.356 | 5 |
| rtdl_FTTransformer | 0.838 | 9.782 | 0.198 | 0.786 | 0.273 | 8 |
| NODE | 0.838 | 9.335 | 0.224 | 0.695 | 0.357 | 3 |
| SVM | 0.835 | 9.847 | 0.121 | 0.812 | 0.318 | 12 |
| DecisionTree | 0.823 | 12.192 | -0.308 | 1.215 | -0.038 | 4 |
| rtdl_MLP | 0.813 | 10.825 | -0.019 | 0.907 | 0.223 | 1 |
| STG | 0.810 | 12.196 | -0.290 | 1.051 | -0.057 | 5 |
| TabNet | 0.804 | 11.643 | -0.216 | 1.104 | 0.075 | 5 |
| MLP | 0.802 | 12.220 | -0.232 | 0.845 | -0.105 | 1 |
| LinearModel | 0.793 | 13.069 | -0.520 | 1.233 | -0.345 | 5 |
| KNN | 0.781 | 14.101 | -0.727 | 1.187 | -0.608 | 0 |
| VIME | 0.756 | 14.711 | -0.849 | 1.192 | -0.694 | 5 |

Table 2: **TuneTables-medium and TuneTables-light are substantially faster with only a modest decrease in accuracy.** We compare the average accuracy and runtime (in seconds) of three versions of TuneTables and find that the medium and light versions of the algorithm are substantially faster on large datasets; we also find that TuneTables-medium sacrifices little accuracy.

| | TuneTables | TuneTables-medium | TuneTables-light |
|---|---|---|---|
| Avg. acc (LARGESCALETABLES , size < 50K) | 0.831 | 0.830 | 0.810 |
| Avg. runtime (LARGESCALETABLES , size < 50K) | 1325 | 1026 | 450 |
| Avg. acc (LARGESCALETABLES , all datasets) | 0.830 | 0.828 | 0.787 |
| Avg. runtime (LARGESCALETABLES , all datasets) | 6975 | 1908 | 486 |
| Avg. acc (TabZilla, all datasets) | 0.861 | 0.855 | 0.854 |
| Avg. runtime (TabZilla, all datasets) | 573 | 305 | 196 |

on a subset of the available validation set when the validation set is large, employs lower patience for early stopping, omits a zero-shot TabPFN grid search over 30 random seeds which TuneTables-standard uses to find more optimal feature selection subsets, and, most impactfully, omits ensembling for datasets larger than a cutoff hyperparameter (which we fix at 150 000 samples). *TuneTables-light* includes all of the efficiency modifications of TuneTables-medium, trains for just one epoch, and uses TabPFN zero-shot to preselect the feature selection method rather than performing a grid search using TuneTables itself. In Table 2, we compare these lighter methods to TuneTables-standard on the LARGESCALETABLES benchmark. TuneTables-medium decreases runtime by 72% compared to TuneTables-standard, while the accuracy decreases by less than a quarter of a percent. Furthermore, the runtime of TuneTables-light is comparable to CatBoost; however, performance also degrades by 5% when going from TuneTables-medium to TuneTables-light. Still, TuneTables-light shows stronger performance than TabPFNs3000 (78.7% vs. 78.1% accuracy) while having a lower inference time.

**Ablations.** In order to better understand the significance of the changes we introduce in our method, we separately ablate the tuned prompt size and the use of ensembles in Table 12 and Table 13, finding that smaller datasets pair well with smaller prompts and rarely benefit from ensembling.

We also compare TuneTables with and without keeping the real data as additional context (referred to as C and NC, respectively, in Section 3); see Appendix Table 12. We find that smaller datasets cannot always be fully learned by the tuned prompt, but for larger datasets, the tuned prompt alone suffices, and in some cases even outperforms the real data.

Table 3: **TuneTables significantly improves accuracy and demographic parity.** In these multi-objective optimization experiments, we consider prompt tuning for mitigating predictive bias, comparing TabPFN to TuneTables, tuning for accuracy alone vs. accuracy and demographic parity. TuneTables improves over TabPFN with respect to both objectives.

| | Adult | | Speeddating | | Compas | | NLSY | | Average | |
|---|---|---|---|---|---|---|---|---|---|---|
| | Acc ↑ | DP ↓ | Acc ↑ | DP ↓ | Acc ↑ | DP ↓ | Acc ↑ | DP ↓ | Acc ↑ | DP ↓ |
| TabPFN | 0.832 | 0.174 | 0.86 | 0.012 | 0.688 | 0.22 | **0.986** | 0.326 | 0.842 | 0.183 |
| TuneTables (Acc) | **0.845** | 0.13 | **0.865** | 0.006 | 0.688 | 0.209 | 0.974 | 0.302 | **0.843** | 0.162 |
| TuneTables (Acc + DP) | 0.837 | **0.034** | 0.863 | **0.003** | **0.693** | **0.121** | 0.965 | **0.277** | 0.840 | **0.109** |

**Sketching and feature selection**   Finally, in Appendix Table 6 we give a study on the three sketching and three feature selection techniques described in Section 2. As described earlier, TabPFNs3000's performance when relying on feature selection plateaus well before approaching parity with CatBoost, a top-performing GBDT, on seven very large datasets.

## 5   TuneTables Extensions

**Mitigating bias with prompt tuning.** Many real-world applications of machine learning involve a set of protected attributes (such as race or gender) that partition the dataset into groups, in which some have higher model performance than others. Removing the sensitive attributes does not fix the algorithmic bias, because the sensitive attributes are often non-trivially correlated with other attributes in the dataset. Due to this issue, researchers have put in significant effort into mitigating the bias of ML models, with the majority of techniques devising new training strategies [9, 53].

Given TabPFN's pretrained nature and considerable retraining cost, the only options for mitigating biased predictions are to run a post-processing routine on the output predictions, which generally do not perform as well as in-processing strategies [66]. We show how to use prompt tuning to substantially reduce the bias of predictions while also improving accuracy.

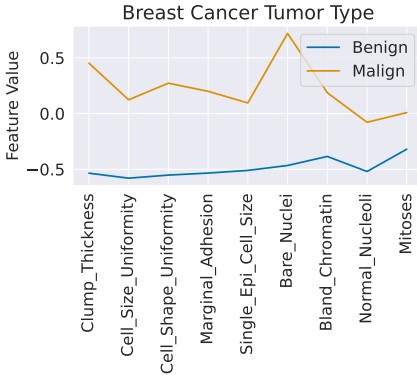

We conduct experiments on four datasets widely used for research in fairness: the Adult Census Income database (with sex as the sensitive attribute) [8], speed dating (with same race as the sensitive attribute) [83], COMPAS (with sex as the sensitive attribute) [5], and National Longitudinal Survey (with gender as the sensitive attribute) [73]. To quantify bias, we use *demographic parity* [25, 76], which measures the difference in probability of a positive outcome among the protected and unprotected groups. Formally, given protected group $G_1$, unprotected group $G_0$, and protected attribute $x_{\cdot,a}$, it can be computed as

Figure 4: **Dataset with high accuracies from just two datapoints**. Shown is a two-example prompt dataset for the breast cancer dataset [78]. Malign class example has higher values for all features than benign class.

$$P_{(x_i,y_i)\in G_0}(y_i = 1 \mid x_{i,a}) - P_{(x_i,y_i)\in G_1}(y_i = 1 \mid x_{i,a}).$$

During prompt tuning, we employ a demographic parity regularizer that aims to minimize the difference in positive outcome probabilities between the two groups:

$$\left| \sum_{(x_i,y_i)\in G_0} P(y_i = 1 \mid x_{i,a}) - \sum_{(x_i,y_i)\in G_1} P(y_i = 1 \mid x_{i,a}) \right|$$

We use the same experimental setup as in Section 4, except that we report one shift rather than the average of three, and TuneTables is fitted to a single prompt rather than ensembled. We compare the default TabPFN to TuneTables, fine-tuning for accuracy alone vs. accuracy and demographic

parity. The latter significantly improves demographic parity, compared to the default TabPFN and TuneTables fine-tuned for accuracy, and enhances accuracy relative to the default TabPFN across all datasets but one; see Table 3.

**Summarizing and understanding datasets with prompt tuning.** While we have demonstrated that TuneTables scales and improves the performance of PFNs, now we show that it can also help understand the discriminative features in a dataset. Often, besides a good predictive model for a dataset, users want to gain further insights into the dataset. In prompt tuning, the tuned smaller dataset can be seen as a summary of the complete dataset that emphasizes discriminative features for the given task. As an example, in Figure 4 we show that on the Breast Cancer dataset [78], a prompt with just two synthetic examples is enough to reach high accuracies and at the same time allows an understanding of the predictive features. For example, in the Breast Cancer dataset, the malign example has higher values for all features compared to the benign example, suggesting high feature values indicate malignancy. We give further examples in Appendix F.

# 6 Related work

**Neural Processes and Prior-Data Fitted Networks.** Prior-data Fitted Networks (PFNs) [35, 55] are a recently-proposed paradigm for machine learning, which show that fast approximate Bayesian inference is possible by training a neural network to mimic the posterior predictive distribution (PPD) in a single forward pass using in-context learning [54, 55, 58]. PFNs were shown to yield state-of-the-art empirical performance on small tabular datasets [35, 51]. PFNs have been used in other applications, including Bayesian optimization [54] learning curve extrapolation [1], and as foundation models for hypernetworks [57].

PFNs are neural processes (NPs) [55]. Recent advances in NP are similar in nature to our work, especially recent works that aim to scale attention-based methods to larger contexts. Feng et al. [27] propose Latent Bottlenecked Attentive Neural Processes (LBANPs), a transformer-based neural process which overcomes the quadratic complexity of transformers by encoding the context dataset into a constant number of latent vectors. Guo et al. [34] propose Versatile Neural Processes (VNP), which increases the capability of NPs to handle compex signals by using a new bottleneck encoder. [63] introduces semi-parametric inducing point networks (SPIN), which can attend to a training set at inference time with linear complexity via inducing point methods. [28] introduce Constant Memory Attention Block (CMAB), an attention block that is permutation-invariant and has constant memory complexity when computing the output, as well as Constant Memory Attentive Neural Processes (CMANPs), a NP that requires constant memory. For additional related work, see Appendix B.

# 7 Conclusions, limitations, and future work

In this work, we gave the first investigation into context optimization techniques for PFNs, allowing us to substantially improve their performance when scaled to large datasets. In particular, we introduced TuneTables, which uses a novel prompt-tuning technique to achieve strong performance on large datasets. We demonstrate that TuneTables mitigates the constraints of TabPFN on the dataset size, number of features, and number of class labels. Additionally, we use prompt tuning to mitigate bias without retraining TabPFN and as an interpretability tool. We open-source our code, results, and all other materials needed to reproduce our work. As PFN models improve, the context optimization techniques explored in our work will allow researchers to further optimize and scale. For example, a next-generation TabPFN might have a longer total context length, and prompt tuning will allow us to push the dataset size even further.

**Limitations.** No current TuneTables is on par with GBDTs in terms of both accuracy and runtime simultaneously. We therefore emphasize that while we achieve strong performance metrics, we do not claim practical superiority of our method over gradient boosting, when taking into account training time. However, given the novelty of our method, we expect future research to further improve the accuracy-runtime tradeoff of TuneTables. It also does not improve on TabPFN for small datasets (fewer than 1000 samples, 100 features and 10 classes); we postulate that this is a result of overfitting.

**Future work.** Parameter-efficient fine-tuning can be used with PFNs to ensure high-quality, differentially private predictions, given that the pretraining is done on purely synthetic data, and prompt

tuning only updates a small number of parameters. Low-rank adaptation (LoRA) [37] and quantized low-rank adaptation (QLoRA) [23] have been used successfully for large language models and would be a promising technique for parameter-efficient fine-tuning of PFNs. Designing a sparse mixture-of-experts PFN using router networks is another promising technique, due to its success in the field of large language models [39].

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

## A Broader Societal Impact Statement

The goal of our work is to investigate context optimization strategies for PFNs, such as prompt-tuning and sketching, which we use to scale and improve the performance of TabPFN. We do not see any negative broader societal impacts of our work that do not already exist in other classification methods. In fact, our work may further facilitate the adoption of TabPFN, which has the benefit of being pretrained and therefore having a lower carbon footprint compared to most deep learning approaches that must be trained from scratch.

Furthermore, we demonstrate that our prompt-tuning strategy makes it possible to mitigate the bias of TabPFN while only fine-tuning a small set of embedding vectors; the authors of TabPFN mentioned that it is critical to study and improve TabPFN under the lens of algorithmic fairness and other dimensions of trustworthy AI [35]. This may allow practitioners to use TabPFN in sensitive settings for the first time, in which the original TabPFN would lead to biased preditions. Overall, our hope is that our work will have a positive impact for both practitioners and researchers, by facilitating the adoption of a model with a low carbon footprint, and by providing the tools to mitigate the bias of PFNs. Likewise, we hope that the possibility to summarize datasets with prompt tuning will add to the toolbox of machine learning practitioners aiming to analyze and interpret their data better, and therefore may have a positive impact on the trustworthieness of machine learning.

## B Additional Related Work

**Tabular classification.** Tabular datasets are the oldest and among the most widely used dataset types in machine learning [10, 67]. GBDTs [29] build an ensemble of decision trees, with each tree fitting the residual of the loss from the previous tree. XGBoost [16] and CatBoost [61] are two of the most widely-used and highest-performing GBDTs. Researchers have also explored many methods based on neural nets [31, 38, 42].

There is an active debate in the community on which family of methods perform best on tabular data: neural nets [6, 42, 47, 60, 65] or GBDTs [10, 31, 32, 67], with the exception of small datasets, on which TabPFN performs the best [35, 51]. Finally, sketching and feature selection methods have been extensively studied in prior works [13, 56].

**Neural Processes and Prior-Data Fitted Networks.** Prior-data Fitted Networks (PFNs) [35, 55] are a recently-proposed paradigm for machine learning, which show that fast approximate Bayesian inference is possible by training a neural network to mimic the posterior predictive distribution (PPD) in a single forward pass using in-context learning [54, 55, 58]. PFNs were shown to yield state-of-the-art empirical performance on small tabular datasets [35, 51]. PFNs have been used in other applications, including Bayesian optimization [54], forecasting [24], and learning curve extrapolation [1].

PFNs are neural processes (NPs) [55]. Recent advances in NP are similar in nature to our work, especially recent works that aim to scale attention-based methods to larger contexts. Feng et al. [27] propose Latent Bottlenecked Attentive Neural Processes (LBANPs), a transformer-based neural process which overcomes the quadratic complexity of transformers by encoding the context dataset into a constant number of latent vectors. Guo et al. [34] propose Versatile Neural Processes (VNP), which increases the capability of NPs to handle compex signals by using a new bottleneck encoder. [63] introduces semi-parametric inducing point networks (SPIN), which can attend to a training set at inference time with linear complexity via inducing point methods. [28] introduce Constant Memory Attention Block (CMAB), an attention block that is permutation-invariant and has constant memory complexity when computing the output, as well as Constant Memory Attentive Neural Processes (CMANPs), a NP that requires constant memory.

For additional related work, see Appendix B.

**Prompt tuning.** Researchers have extensively studied prompting techniques for large language models (LLMs) [49]. In such methods, one modifies the input into a *prompt*, and the LLM predicts the subsequent tokens, from which it derives the output predictions. *Prompt tuning* techniques typically start with a pretrained LLM and use the training data of a downstream task to find the best prompt for this task. 'Hard' prompt tuning involves finding the best natural language prompts using discrete

search techniques [40, 77], while 'soft' prompt tuning optimizes a prompt directly in the embedding space of the model [46, 84]. Soft prompt tuning has also been applied to multi-modal models such as vision-language models [70, 85, 86].

# C  Additional experimental details

**Hyperparameter settings for baselines.** For the baselines, we use the same hyperparameter search space as in prior work [51] (which is itself similar to other works [42]). We choose the best configuration among the default parameter sets and the additional 29 sets, using k-fold cross-validation with 10 folds. We follow the best configurations of [51] with exceptions; for CatBoost and XGBoost, we increase the range of tree depth by a factor of 10, since a high depth is often more appropriate for very large datasets. We find that this improves performance on large datasets, but sometimes harms performance on small datasets, and increases the runtime – see Appendix G for the raw results. For each dataset, each algorithm is allowed a maximum runtime of 10 hours for the entire search process. For ExcelFormer, we decrease the number of dimensions and depth of the default transformer, as we find that without this modification, it times out on even medium-sized datasets.

**Hyperparameter settings for TuneTables.** Consistent with the experimental design in [51], we perform hyperparameter optimization of TuneTables for up to 30 variants; however, we utilize a grid search rather than Optuna. We use a batch size of 1 and no gradient aggregation. The space over which we conduct grid search in TuneTables is conditioned on metadata; number of features, number of samples, number of classes. The feature and class splits are set by the limits of the particular TabPFN checkpoint we optimize. When we reach a leaf node, TuneTables-standard and TuneTables-medium conduct a grid search over a fixed range of configurations. The size of our search space is always < 30, and usually < 10. For feature-large datasets, TuneTables-light conducts additional optimization in the feature space prior to grid search. Further details on our search algorithm can be found in the repository associated with the paper. TuneTables hyperparameter settings can be found in chart form in Table 5.

## C.1  Memory Efficiency

In our experiments, we find that we exceed hardware limitations around 3000 samples with a batch size of 1 when using TabPFN on a GPU with 48GB of VRAM. While FlashAttention, a popular optimization, is memory-linear in sequence length, but it has not yet been implemented in the public release of TabPFN, which relies on an older version of PyTorch. Flash attention may be integrated into a future release, in which case any benefits will carry over to our method. Even so, inference time continues to be quadratic, and there are other overheads on GPU memory. Overall, there remains a considerable need for new algorithmic methods which can be scaled up to the sequence lengths required by large tabular datasets.

## C.2  Feature selection and sketching

We ablate strategies for both sketching (subsampling) and feature selection for scaling TabPFN. We consider three sketching methods (*random*, $k$-*means*, and *CoreSet*) and three feature selection methods (*random*, *PCA*, *mutual information*) as described in Section 2. In addition to sketching, we consider two different methods for sampling class labels: *equal* and *proportional*.

For efficient coreset selection, we use a variant of Farthest Point Sampling [26]; after selecting an initial set of n=5 random points, we compute the distance of each point in the dataset to the set of already selected points. Next, we add to the set of selected points the point whose distance to the selected points is maximal. Finally, we update the distances of all the points according to the updated selected set; and continue iteratively.

We limit our algorithmic comparison to TabPFN to CatBoost, which is the overall best-performing model in [51]. We compare all combinations of sketching and feature selection with both CatBoost and TabPFN; see Table 6. Interestingly, we find that *random* sketching performs just as well as the more involved algorithms, $k$-means and *CoreSet*. On the other hand, PCA significantly outperforms mutual information and random feature selection methods, when the original number of features is large.

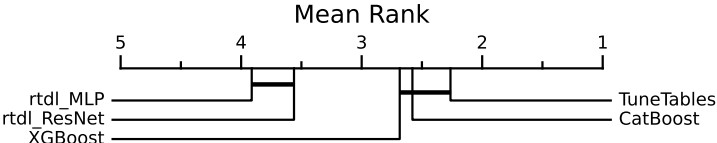

Figure 5: **TuneTables is competitive with state-of-the-art tabular models.** A critical difference plot according to mean accuracy rank across all LARGESCALETABLES datasets with fewer than 50 000 samples. Algorithms which are *not significantly different* ($p > 0.05$) are connected with a horizontal black bar. TuneTables achieves the highest mean rank of any algorithm. This plot is similar to Figure 2, but the search spaces for XGBoost and CatBoost are expanded to include more trees.

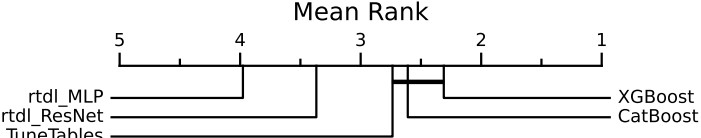

Figure 6: We present a critical difference plot according to mean accuracy rank on LARGESCALETA-BLES . Algorithms which are *not significantly different* ($p > 0.05$) are connected with a horizontal black bar. Across these datasets, TuneTables achieves performance that is not significantly different from CatBoost or XGBoost.

## C.3 Additional results

First, we present the large class table from Section 4. See Table 7.

In Table 1, we gave aggregate statistics for TuneTables and baselines across all datasets of size less than 50 000. Now, in Table 8, we present results for TuneTables, CatBoost, and XGBoost on all datasets individually, including datasets of size nearly two million. Note that we report the end-to-end runtime of all methods, including 30 iterations of search (for example, TabPFNs3000 takes about $30\times$ more time than TabPFN as reported in prior work [51]). In Figure 6, we present a critical difference diagram similar to Figure 2 but over all 29 datasets. Across these 29 datasets, TuneTables achieves performance that is not statistically different from CatBoost or XGBoost.

**Neural net comparison.** In Table 1 and Table 8, we compared TuneTables to GBDTs and two other neural nets. Now, we compare TuneTables to two additional neural nets: MLP and TabTransformer [38] (for four total neural net baselines). Since transformer-based methods have a higher memory consumption, we obtained full results on 17 total datasets. See Table 9. We find that TuneTables substantially outperforms all other neural nets on average across all datasets, achieving the lowest average rank at 1.93, and achieving the highest average Z-Score of 0.85, far above the second-place neural net's value of 0.18.

**Inference Time** We give the full details for inference time; see Table 10.

**Full results for TuneTables-medium and TuneTables-light** Recall at the end of Section 4, we introduced TuneTables-medium and TuneTables-light, showing that they can substantially decrease the runtime of the standard TuneTables with just a small reduction in accuracy. While we presented summary results in Table 2, now we present the full results in Table 11. For descriptions of TuneTables variants, see Section 4.

## C.4 Ablation study

We report our results ablating the core components of our methods: the prompt length, its use during training and inference, the use of ensembles, and the prompt tuning procedure itself (against regular finetuning). Beginning with the prompt length, we see in Table 12 that while some dataset results are not sensitive to the prompt length, others vary significantly, and generally enjoy longer prompts. Ablating the importance of having real-data context during inference (same table) we find that it is

important for smaller datasets. On the larger datasets, the variant with no real data at all is better in specific cases. However, having no such data during inference, a model would perform better not having it during training as well. Introducing real data first during inference is usually harmless, but significantly deteriorates the results on specific cases. Fine tuning the entire model is not only significantly more memory-intensive but also underperforms TuneTables in terms of accuracy.

In Table 13 we show the results of our method with or without ensembles, and with or without context (in training and inference). Our ensembling strategy is beneficial in both cases. When using ensembles, the results with or without context are generally similar, although one variant may outperform the other on specific datasets.

## D   TuneTables additional details

**Tuned prompt implementation details.**   We implement our tuned prompts by prepending randomly initialized vectors of identical dimension to the TabPFN encoder to the real data points fed to TabPFN as context during training.

In the **CT** ('context in training') setting, we continue to provide real data points as part of the training context; the quantity of real data provided for each batch is drawn with uniform probability from a random variable which ranges from zero to a fixed upper bound, which is passed to the model as a hyperparameter. Usually that upper bound is 1152, the default setting for TabPFN, unless the dataset is extremely small, in which case we select it according to the limitations of that dataset.

In the **NCT** ('no context in training') setting, no real data is provided as part of the training context; all data in the training set is used to fit the randomly initialized prompt. There are always a fixed number of data points provided, and it is the same for every batch. Usually that number is 128, unless the dataset is extremely small, in which case we select it according to the limitations of that dataset.

We note that it is also possible to evaluate models trained in either setting either with context (C) or without (NC). In our experiments, we always evaluate all models on both, and report the setting with higher validation accuracy at test time.

We find that on smaller datasets, the setting with context (C) sometimes outperforms the setting without context (NC), even after the prompt has been fitted, perhaps because the tuned prompt overfits on a small amount of available training data. However, when using ensembling, for datasets with more than 3000 samples in the training set, the NC setting is as good or better than the C setting. See Table 13. We leave further investigation of this phenomenon to future work.

**Loss function.**   Most TuneTables experiments are optimized using the cross-entropy loss between the labels assigned to the training data (P) and TuneTables's outputs (Q):

$$H(P, Q) = -\sum_x P(x) \log Q(x) \tag{5}$$

To reduce the effects of overfitting on datasets with fewer samples, we extend our grid search to include the choice of loss function. We find that many such datasets benefit from fitting the tuned prompt via the KL divergence loss between the PFN's outputs (P) and TuneTables (Q).

$$KL(P||Q) = \sum_x P(x) \log \left( \frac{P(x)}{Q(x)} \right) \tag{6}$$

**Compute.**   We conduct all of our prompt tuning experiments on Google Cloud Platform (GCP), using a single NVIDIA L4 TPU with 24GB VRAM for each experiment.

**Size of tuned prompts.**   We ablate the size of tuned prompts in Table 12 and find that the larger tuned prompts generally perform better on datasets with more samples, while smaller tuned prompts excel on datasets with few samples. For this reason, the experimental results in Section 4 are reported on prompts of size 10 or 1000, conditioned on the number of samples in the input dataset.

**Duration of training.**     Our experiments run for up to 100 epochs. We employ early stopping if our key metric (Accuracy) fails to improve after a fixed number of epochs.

**Evaluation of tuned prompts.**     We validate our tuned prompts every epoch on a subset of the entire validation set if the validation set is large. At a fixed interval, we run the entire validation and test sets. After the experiment concludes, we report results from the epoch with the best key metric score.

**Fine-tuned setting.**     In the fine-tuned setting, our parameters are the same, except that we use a lower fixed learning rate of $1e - 3$.

**Ensembling over tuned prompts.**     While individual tuned prompts are already a substantial improvement on TabPFN for sample-large datasets, we find that these improvements sometimes compound when we ensemble over multiple tuned prompts.

We draw the inspiration for our ensembling approach from the internal ensembling used in TabPFN, which averages predictions over permutations of feature indices and label indices[35]. For the results presented in Section 4, we ensemble over ten permutations of each dataset, averaging the top two ensemble member predictions (as measured by NC accuracy on the validation set in the NCT setting, or C accuracy in the CT setting).

In a TuneTables ensemble, each ensemble member fits its own tuned prompt to the data. Variance in the ensemble members is introduced by differences in the random initialization of the tuned prompt, as well as permuting the order of features and labels, a la TabPFN, one time before each tuned prompt is fitted.

See Table 13 for ablation studies of the effectiveness of ensembling.

# E    Regression experiments

In this section, we show that using prompt tuning, we are also able to extend PFNs to perform well on regression datasets, further highlighting the flexibility of our approach.

## E.1    Adapting TuneTables to regression problems.

Adapting TuneTables to regression problems introduces new challenges, as TabPFN was not designed for use with such problems. In particular, we find that end-to-end fine-tuning outperforms prompt tuning in this setting, and that the most effective grid search for regression problems utilizes a space of *PFN base models*. Put another way, TuneTables-regression does not use prompt tuning, and does not always use TabPFN as its foundation model. Rather, it searches a space of foundation models to find the best performer for a particular regression problem. We search over three foundation models; TabPFN, the checkpoint released in [55] and a new PFN we train from scratch for 10 epochs on a synthetic prior of regression-specific datasets.

## E.2    Details on the datasets used for the experiments and baseline experimental design

Similar to [51], each algorithm is tuned for each dataset by maximizing the R-squared (R2) metric. Each dataset corresponds to an OpenML task, and can be preprocessed exactly like the classification datasets used in other experiments. The 15  datasets used in these experiments are "Bank-Note-Authentication- UCI" (OpenML task 361002), "EgyptianSkulls" (5040), "Wine" (190420), "Wisconsin-breast- cancer-cytology-features" (361003), "bodyfat" (5514), "california" (361089), "chscase-foot" (5012), "cleveland" (2285), "colleges" (359942), "cpu-small" (4883), "liver-disorders" (52948), "meta" (4729), "mv" (4774), "pbc" (4850), and "veteran" (4828). Unlike prior work, we report non-normalized averages, as we believe it gives a more realistic indication of real-world performance at a glance.

We use 12 deep learning and boosted tree algorithms as baselines; VIME, TabTransformer, TabNet, DANet, STG, MLP, LightGBM, CatBoost, XGBoost, NODE, DeepFM.

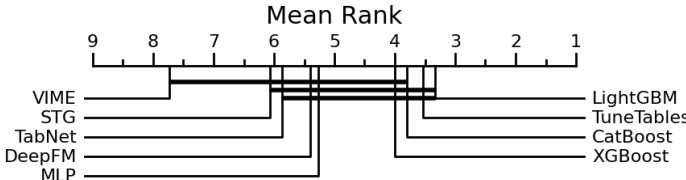

Figure 7: **TuneTables is competitive with state-of-the-art tabular models on regression.**

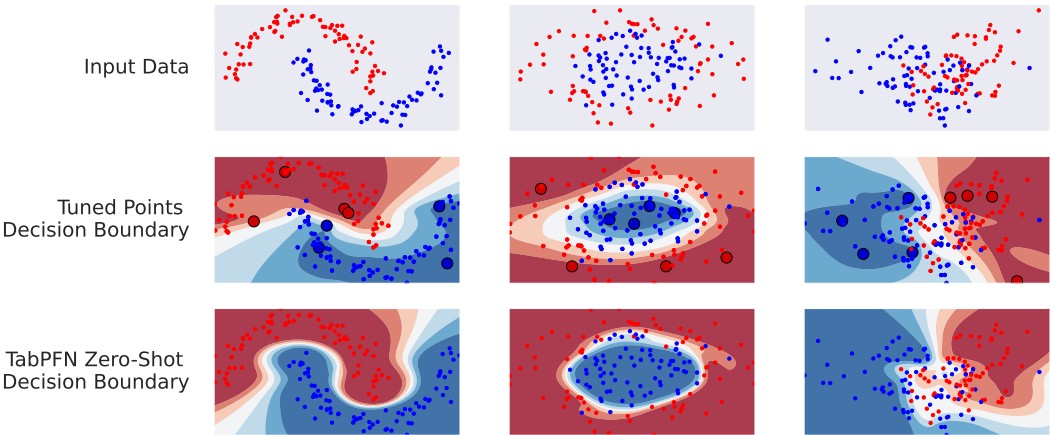

Figure 8: **Decision boundaries on 2D datasets.** Shown are prompt-tuned TabPFN decision boundaries as well as the regular zero-shot TabPFN decision boundaries. Larger dots in middle row represent the tuned points for the two classes.

### E.3 Results of regression experiments

In Table 14 results and Figure 7, we show the results of our experiments. Overall, we see that TuneTables is competitive with top-performing algorithms across a variety of metrics; mean win rank, average $R^2$ score, etc. Equally importantly, TuneTables is significantly stronger on several datasets than both boosted trees and deep baselines.

## F   Summarization details and decision boundaries of prompt-tuned TabPFN on toy 2D Problems

For the experiments with prompts of only two examples, we prompt-tune one example per class for 500 epochs. The prompt is input directly into TabPFN without undergoing any preprocessing steps that the original data went through. Therefore, the tuned feature values do not correspond directly to the original values in the dataset. For the breast cancer dataset, it reaches 100% accuracy; for the diabetes dataset, it reaches 78.7% accuracy.

We also show decision boundaries of TabPFN and prompt-tuned TabPFN on toy 2d classification problems using scikit-learn [59] in Fig. 8. The prompt contains four datapoints for each class. One can see how the points are tuned to recover the decision boundaries. Due to the very low prompt length and the low dataset size, the prompt-tuned decision boundaries are slightly less accurate than TabPFN (consistent with our results for small datasets in Table 12).

## G   Results for Deeper GBDTs

In the results in Section 4, we compare TuneTables to the GBDTs, XGBoost and CatBoost, using a search space in which the number of trees can range from 1 to 1000. Now, we compare TuneTables to XGBoost and CatBoost when they have expanded search spaces, where the number of trees can

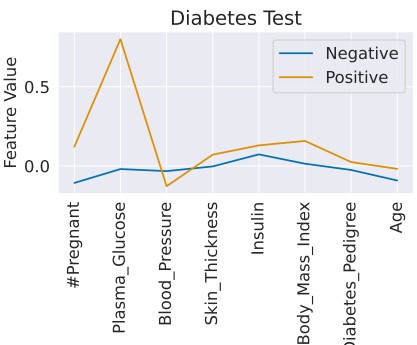

Figure 9: **Diabetes dataset [68] with high accuracies from just two samples.**

range from 1 to 10 000. We re-compute the critical difference plot and table results: see Figure 5, Table 15, and Table 16.

In Figure 5 and Table 15, we see that TuneTables still performs favorably compared to CatBoost and XGBoost on datasets of size up to 50 000. Note that the larger search space may perform better on larger datasets, it is also more challenging to search through in 30 iterations. We present the full results on all datasets in Table 16.

Table 4: List of all datasets used in Section 4 and Appendix C. LARGESCALETABLES datasets are listed in part one. Datasets in part two were used for the class extension experiments in Table 7. Datasets in part three were used for fairness experiments in Section 5. The additional dataset in part four was used for intepretability experiments in Section 5. Datasets in part 5 were used for the ablations on sketching and feature selection in Appendix C. For the list of 98 used in Table 8, see https://github.com/penfever/TuneTables/blob/main/datasets-used.csv.

| Dataset | num. classes | num. features | num. samples |
| --- | --- | --- | --- |
| click-prediction-small | 2 | 11 | 1997410 |
| poker-hand | 10 | 10 | 1025009 |
| agrawal1 | 2 | 9 | 1000000 |
| BNG (labor) | 2 | 16 | 1000000 |
| airlines | 2 | 7 | 539383 |
| albert | 2 | 78 | 425240 |
| BNG (vote) | 2 | 16 | 131072 |
| connect-4 | 2 | 28 | 98050 |
| higgs | 3 | 42 | 67557 |
| volkert | 10 | 180 | 58310 |
| riccardo | 2 | 4296 | 20000 |
| elevators | 2 | 18 | 16599 |
| eeg-eye-state | 2 | 14 | 14980 |
| har | 6 | 561 | 10299 |
| dilbert | 5 | 2000 | 10000 |
| robert | 10 | 7200 | 10000 |
| christine | 2 | 1636 | 5418 |
| bioresponse | 2 | 1776 | 3151 |
| kc-1 | 2 | 21 | 2109 |
| car | 4 | 6 | 1728 |
| cmc | 3 | 9 | 1473 |
| blood-transfusion | 2 | 4 | 748 |
| balance-scale | 3 | 4 | 625 |
| climate | 2 | 18 | 540 |
| cylinder-bands | 2 | 37 | 540 |
| dresses-sales | 2 | 12 | 500 |
| colic | 2 | 22 | 368 |
| ecoli | 8 | 7 | 336 |
| heart-c | 2 | 13 | 303 |
| breast-cancer | 2 | 9 | 286 |
| openml__ldpa__9974 | 11 | 7 | 164860 |
| openml__walking-activity__9945 | 22 | 4 | 149332 |
| openml__Devnagari-Script__167121 | 46 | 1024 | 92000 |
| openml__helena__168329 | 100 | 27 | 65196 |
| openml__chess__3952 | 18 | 6 | 28056 |
| openml__kropt__2076 | 18 | 6 | 28056 |
| openml__letter__6 | 26 | 16 | 20000 |
| openml__isolet__3481 | 26 | 617 | 7797 |
| openml__texture__125922 | 11 | 40 | 5500 |
| openml__one-hundred-plants-texture__9956 | 100 | 64 | 1599 |
| openml__vowel__3022 | 11 | 12 | 990 |
| openml__soybean__41 | 19 | 35 | 683 |
| openml__collins__3567 | 15 | 21 | 500 |
| openml__arrhythmia__5 | 13 | 279 | 452 |
| openml__primary-tumor__146032 | 21 | 17 | 339 |
| adult-census | 2 | 15 | 32561 |
| SpeedDating | 2 | 121 | 8378 |
| compas-two-years | 2 | 12 | 4966 |
| nlsy (national-longitudinal-survey-binary) | 2 | 17 | 4908 |
| Diabetes | 322 | 10 | 442 |
| skin-segmentation | 2 | 3 | 245057 |
| FM (Fashion-MNIST) | 10 | 784 | 70000 |
| CIFAR-10 | 10 | 3072 | 60000 |
| gddc (gas-drift-different-concentrations) | 6 | 129 | 13910 |
| pendigits | 10 | 16 | 10992 |
| mfeat-factors | 10 | 216 | 2000 |
| mfeat-pixel | 10 | 240 | 2000 |
| semeion | 10 | 256 | 1593 |
| hill-valley | 2 | 100 | 1212 |

Table 5: TuneTables and TabPFNs3000 hyperparameter configurations based on number of samples.

| Parameter | TabPFN | TuneTables ≤ 2000 | TuneTables > 2000 |
|---|---|---|---|
| Batch Size | 1 | - | - |
| Real Data Qty | 3000 | 0 | {1152, 0} |
| Ensemble Size | 32 | 1 | 10 |
| Tuned Prompt Dimension | - | 10 | 1000 |
| Ensemble Method | - | - | Avg Top 2 |
| Epochs | - | {7, 60} | 100 |
| Warmup | - | 10 | 10 |
| Sequence Length Per Batch | - | Fixed | {Variable, Fixed} |
| Early Stopping | - | {2, 6} | 6 |
| Learning Rate | - | 3e-2 | 1e-3 |
| Validation Frequency | - | 2 | 2 |
| Max Val Size During Training | - | 2000 | 2000 |
| Optimizer | - | AdamW, Default Settings, Weight Decay 0 | AdamW, Default Settings, Weight Decay 0 |
| Loss | - | {Cross-Entropy, KL Divergence} | - |
| Tuned Prompt Labels | - | {Equal, Proportional} | - |

Table 6: **Comparative performance of TabPFN and CatBoost with sketching, feature selection, and sampling methods.** On a distinct subset of the datasets in [51] selected to emphasize datasets with many features or many samples, we compare CatBoost and TabPFNs3000. When both models are limited to 3000 samples, TabPFNs3000 performs better on 12 of 17 datasets where significant differences exist. When CatBoost is allowed access to the entire training data, the win rate is identical. In most cases, random sample selection is sufficient for optimal performance. Both models benefit from PCA and mutual information dimension reduction when the feature space is large. The columns labeled SKT / FTS / SMP list the best performing method for sketching, feature subsampling and label-aware sketching technique, respectively. Label-aware sketching refers to a strategy where we either sample instances proportionate to their labels, or we oversample minority classes with replacement to create a class-balanced distribution. While the choice of label-aware sketching strategy is often impactful (and we use it in TuneTables), and the choice of feature subselection method can be important for some datasets, in all but one case, no sketching method we test outperforms random sampling. **Bold** indicates the best-performing model(s).

| | Full CatBoost | Best CatBoost | Random CatBoost | Best TabPFN | Random TabPFN | SKT / FTS / SMP CatBoost | SKT / FTS / SMP TabPFN |
|---|---|---|---|---|---|---|---|
| airlines_189354 | **0.653** | 0.637 | 0.637 | 0.594 | 0.589 | RND / RND / PR | RND / RND / PR |
| albert_189356 | **0.698** | 0.657 | 0.657 | 0.64 | 0.64 | RND / RND / PR | RND / RND / PR |
| CIFAR_10_167124 | **0.434** | 0.37 | 0.342 | 0.373 | 0.372 | RND / PCA / PR | RND / RND / PR |
| connect-4_146195 | **0.749** | 0.716 | 0.716 | 0.66 | 0.659 | RND / RND / PR | RND / RND / PR |
| eeg-eye-state_14951 | 0.832 | 0.808 | 0.806 | **0.932** | **0.932** | RND / RND / PR | RND / RND / EQ |
| elevators_3711 | 0.855 | 0.838 | 0.838 | **0.9** | 0.899 | RND / MUT / PR | RND / RND / PR |
| FM_146825 | **0.843** | 0.787 | 0.787 | **0.835** | 0.812 | RND / RND / PR | RND / PCA / PR |
| gddc_9987 | 0.97 | 0.976 | 0.955 | **0.994** | 0.993 | RND / PCA / EQ | RND / RND / PR |
| higgs_146606 | **0.71** | 0.684 | 0.684 | 0.665 | 0.661 | RND / RND / PR | RND / RND / PR |
| hill-valley_145847 | 0.514 | 0.514 | 0.514 | **0.56** | **0.56** | RND / RND / PR | RND / RND / PR |
| mfeat-factors_12 | 0.954 | 0.95 | 0.943 | **0.973** | **0.973** | KMN / RND / EQ | RND / RND / PR |
| mfeat-pixel_146824 | 0.955 | 0.951 | 0.951 | **0.971** | 0.97 | RND / RND / PR | RND / RND / PR |
| pendigits_32 | 0.972 | 0.966 | 0.964 | **0.995** | 0.993 | RND / RND / PR | RND / RND / PR |
| poker-hand_9890 | **0.664** | 0.572 | 0.561 | 0.519 | 0.515 | RND / RND / PR | RND / RND / PR |
| riccardo_168338 | 0.951 | 0.956 | 0.93 | **0.991** | 0.982 | RND / PCA / EQ | RND / MUT / EQ |
| robert_168332 | **0.446** | 0.367 | 0.367 | 0.384 | 0.359 | RND / RND / PR | RND / PCA / EQ |
| semeion_9964 | 0.887 | 0.869 | 0.863 | **0.915** | **0.915** | RND / MUT / EQ | RND / RND / PR |
| ss_9965 | **0.994** | 0.989 | 0.987 | **0.999** | **0.999** | RND / RND / PR | RND / RND / PR |
| volkert_168331 | **0.608** | 0.56 | 0.56 | 0.557 | 0.555 | RND / RND / PR | RND / RND / PR |
| Average | 0.773 | 0.746 | 0.740 | 0.761 | 0.757 | RND / RND / PR | RND / RND / PR |

Table 7: **Comparison of algorithms on datasets with a large number of classes.** TuneTables can effectively handle datasets with more classes than the ones used for pretraining, which was not possible with TabPFN. For each algorithm, we compute its mean test accuracy, and mean rank in terms of accuracy. We also compute the mean Z-score, computed by normalizing the set of results on each dataset (by mean 0 std. 1), so that each dataset has the same weight, and averaging each algorithm's normalized performances. We see that TuneTables performs the best across all performance-oriented metrics. Fractional num. wins values are averaged over three splits per dataset, and reflect the presence of multi-way ties on certain splits.

| Method | Mean Acc. | Mean Rank | Mean Z-Score | Std. Z-Score | Med. Z-Score | Num. Wins |
|---|---|---|---|---|---|---|
| XGBoost | 0.885 | 2.000 | 1.052 | 0.558 | 1.245 | 12.0 |
| TuneTables | 0.779 | 2.524 | 0.602 | 0.932 | 0.971 | 5.0 |
| rtdl_ResNet | 0.776 | 3.929 | 0.053 | 0.826 | -0.006 | 2.0 |
| KNN | 0.789 | 3.976 | 0.121 | 0.262 | 0.103 | 0.0 |
| CatBoost | 0.781 | 4.524 | -0.177 | 0.960 | 0.107 | 2.0 |
| RandomForest | 0.751 | 5.214 | -0.628 | 0.754 | -0.767 | 0.0 |
| rtdl_MLP | 0.624 | 5.833 | -1.022 | 0.908 | -1.257 | 0.0 |

Table 8: **Comparison of the top-performing methods on LARGESCALETABLES .** For each algorithm, we show its test accuracy and end-to-end runtime in seconds. In this table, Table 11 and Table 16, we report the summed runtime for all random seeds (30 in the case of all algorithms except for TuneTables, which generally requires fewer than 30). Since TuneTables uses TabPFN predictions on small datasets, we conservatively report its end to end runtime as the sum of TabPFNs3000 runtime and TuneTables grid search runtime. This prediction is also extremely conservative for TabPFNs3000, as 30 random seeds are unnecessary on datasets with fewer than 100 features. We report runtime in this manner to mitigate concerns that we are advantaging PFNs over boosted trees. We also show the number of samples in each dataset. All datasets above the line were included in Table 1.

| Dataset | Size | TabPFNs3000 | | TuneTables | | CatBoost | | XGBoost | |
|---|---|---|---|---|---|---|---|---|---|
| | | Acc. | Runtime | Acc. | Runtime | Acc. | Runtime | Acc. | Runtime |
| breast-cancer | 286 | 0.765 | 29 | 0.770 | 65 | 0.770 | 37 | 0.724 | 4 |
| heart-c | 303 | 0.848 | 40 | 0.903 | 66 | 0.903 | 21 | 0.839 | 2 |
| ecoli | 336 | 0.848 | 30 | 0.843 | 66 | 0.892 | 73 | 0.833 | 17 |
| colic | 368 | 0.856 | 39 | 0.892 | 66 | 0.874 | 218 | 0.883 | 3 |
| dresses-sales | 500 | 0.578 | 41 | 0.580 | 122 | 0.640 | 40 | 0.640 | 5 |
| cylinder-bands | 540 | 0.800 | 41 | 0.846 | 82 | 0.827 | 94 | 0.864 | 6 |
| climate | 540 | 0.959 | 59 | 0.951 | 97 | 0.963 | 25 | 0.926 | 4 |
| balance-scale | 625 | 0.990 | 29 | 0.995 | 55 | 0.899 | 53 | 0.894 | 145 |
| blood-transfusion | 748 | 0.801 | 25 | 0.782 | 56 | 0.756 | 24 | 0.760 | 58 |
| cmc | 1473 | 0.554 | 91 | 0.556 | 109 | 0.561 | 78 | 0.561 | 13 |
| kc-1 | 2109 | 0.862 | 168 | 0.856 | 187 | 0.856 | 49 | 0.859 | 6 |
| bioresponse | 3151 | 0.797 | 638 | 0.798 | 3012 | 0.788 | 113 | 0.800 | 102 |
| christine | 5418 | 0.742 | 666 | 0.755 | 3920 | 0.736 | 331 | 0.743 | 4 |
| robert | 10000 | 0.250 | 964 | 0.414 | 2397 | 0.464 | 599 | 0.503 | 2118 |
| dilbert | 10000 | 0.922 | 761 | 0.992 | 3749 | 0.949 | 809 | 0.978 | 1144 |
| har | 10299 | 0.936 | 370 | 0.981 | 2657 | 0.985 | 302 | 0.994 | 232 |
| eeg-eye-state | 14980 | 0.940 | 178 | 0.986 | 1929 | 0.907 | 28 | 0.946 | 18 |
| elevators | 16599 | 0.902 | 186 | 0.902 | 1297 | 0.891 | 176 | 0.896 | 234 |
| riccardo | 20000 | 0.922 | 1395 | 0.995 | 5247 | 0.997 | 692 | 0.998 | 735 |
| volkert | 58310 | 0.567 | 459 | 0.693 | 6331 | 0.666 | 230 | 0.703 | 604 |
| higgs | 67557 | 0.671 | 931 | 0.714 | 4084 | 0.724 | 80 | 0.725 | 46 |
| connect-4 | 98050 | 0.668 | 696 | 0.817 | 5395 | 0.807 | 204 | 0.855 | 106 |
| BNG (vote) | 131072 | 0.968 | 1976 | 0.974 | 2493 | 0.975 | 42 | 0.975 | 16 |
| albert | 425240 | 0.642 | 2363 | 0.658 | 17518 | 0.706 | 1078 | 0.690 | 76 |
| airlines | 539383 | 0.600 | 2602 | 0.653 | 44434 | 0.664 | 155 | 0.673 | 184 |
| BNG (labor) | 1000000 | 0.937 | 5518 | 0.967 | 7717 | 0.971 | 226 | 0.970 | 179 |
| agrawal1 | 1000000 | 0.948 | 5158 | 0.950 | 45504 | 0.951 | 142 | 0.951 | 98 |
| poker-hand | 1025009 | 0.531 | 2423 | 1.000 | 10471 | 0.912 | 5331 | 0.940 | 439 |
| click-prediction-small | 1997410 | 0.833 | 10421 | 0.837 | 33148 | 0.842 | 85 | 0.843 | 213 |
| Average | | 0.781 | 1320 | 0.830 | 6975 | 0.823 | 390 | 0.826 | 234 |

Table 9: **Comparison of neural nets on LARGESCALETABLES .** We compare TuneTables to other prominent deep learning methods for tabular data on the 17 datasets in LARGESCALETABLES for which all algorithms reported results. For each algorithm, we compute its different metrics of accuracy and rank. We also compute the mean Z-score, computed by normalizing the set of results on each dataset (by mean 0 std. 1), so that each dataset has the same weight, and averaging each algorithm's normalized performances. We see that TuneTables performs the best across all performance-oriented metrics. Fractional num. wins values are averaged over three splits per dataset, and reflect the presence of multi-way ties on certain splits.

| Method | Mean Acc. | Med Acc. | Mean Rank | Med. Rank | Mean Z-Score | Med. Z-Score | Num. Wins |
|---|---|---|---|---|---|---|---|
| TuneTables | **0.81** | **0.85** | **1.93** | **1.0** | **0.85** | **0.91** | **9.67** |
| rtdl_ResNet | 0.77 | 0.78 | 2.68 | 2.0 | 0.18 | 0.19 | 3.33 |
| TabTransformer | 0.76 | 0.73 | 3.05 | 3.0 | -0.10 | 0.05 | 3.00 |
| rtdl_MLP | 0.73 | 0.76 | 3.55 | 4.0 | -0.34 | -0.55 | 0.83 |
| MLP | 0.72 | 0.74 | 3.79 | 4.0 | -0.59 | -0.55 | 0.17 |

Table 10: **Comparison of inference times across datasets.** In this table, we compare the inference time of TabPFNs3000 (which uses up to 3000 points of real data as context, when available) and four common configurations for TuneTables. We report inference time for the entire test set, as well as per 1000 samples, estimated from the inference time for the entire test set. At inference time, TuneTables does not require ensembling to achieve high accuracy, while TabPFNs3000 does. Therefore, TuneTables is considerably faster.

| Dataset | N. classes | N. Feats | N. Samples | TabPFNs3000 3000 pts, ens. | per1k | TuneTables-pt10-C 3000 pts | per1k | TuneTables-pt10-NC 0 pts | per1k | TuneTables-pt1000-C 3000 pts | per1k | TuneTables-pt1000-NC 0 pts | per1k |
|---|---|---|---|---|---|---|---|---|---|---|---|---|---|
| breast-cancer | 2 | 9 | 286 | 0.951 | 3.325 | 0.103 | 0.36 | 0.103 | 0.36 | 0.119 | 0.416 | 0.109 | 0.381 |
| heart-c | 2 | 13 | 303 | 1.011 | 3.337 | 0.103 | 0.34 | 0.102 | 0.337 | 0.111 | 0.366 | 0.107 | 0.353 |
| ecoli | 8 | 7 | 336 | 0.99 | 2.946 | 0.094 | 0.28 | 0.092 | 0.274 | 0.094 | 0.28 | 0.091 | 0.271 |
| colic | 2 | 22 | 368 | 1.045 | 2.84 | 0.103 | 0.28 | 0.102 | 0.277 | 0.124 | 0.337 | 0.112 | 0.304 |
| dresses-sales | 2 | 12 | 500 | 1.024 | 2.048 | 0.098 | 0.196 | 0.09 | 0.18 | 0.106 | 0.212 | 0.092 | 0.184 |
| climate | 2 | 18 | 540 | 1.067 | 1.976 | 0.098 | 0.181 | 0.09 | 0.167 | 0.11 | 0.204 | 0.095 | 0.176 |
| cylinder-bands | 2 | 37 | 540 | 1.317 | 2.439 | 0.088 | 0.163 | 0.091 | 0.169 | 0.108 | 0.2 | 0.093 | 0.172 |
| balance-scale | 3 | 4 | 625 | 0.7 | 1.12 | 0.115 | 0.184 | 0.105 | 0.168 | 0.147 | 0.235 | 0.104 | 0.166 |
| blood-transfusion | 2 | 4 | 748 | 0.595 | 0.795 | 0.123 | 0.164 | 0.117 | 0.156 | 0.149 | 0.199 | 0.107 | 0.143 |
| cmc | 3 | 9 | 1473 | 1.485 | 1.008 | 0.099 | 0.067 | 0.093 | 0.063 | 0.105 | 0.093 | 0.063 | 0.063 |
| kc-1 | 2 | 21 | 2109 | 2.623 | 1.244 | 0.12 | 0.057 | 0.11 | 0.052 | 0.203 | 0.096 | 0.116 | 0.055 |
| bioresponse | 2 | 1776 | 3151 | 5.623 | 1.785 | 0.23 | 0.073 | 0.116 | 0.037 | 0.338 | 0.107 | 0.114 | 0.036 |
| christine | 2 | 1636 | 5418 | 5.392 | 0.995 | 0.486 | 0.09 | 0.183 | 0.034 | 0.725 | 0.134 | 0.167 | 0.031 |
| dilbert | 5 | 2000 | 10000 | 9.355 | 0.936 | 0.74 | 0.074 | 0.204 | 0.02 | 1.126 | 0.113 | 0.232 | 0.023 |
| robert | 10 | 7200 | 10000 | 9.369 | 0.937 | 0.718 | 0.072 | 0.202 | 0.02 | 1.143 | 0.114 | 0.208 | 0.021 |
| har | 6 | 561 | 10299 | 9.399 | 0.913 | 0.804 | 0.078 | 0.219 | 0.021 | 1.27 | 0.123 | 0.236 | 0.023 |
| eeg-eye-state | 2 | 14 | 14980 | 8.235 | 0.55 | 1.086 | 0.072 | 0.245 | 0.016 | 1.683 | 0.112 | 0.284 | 0.019 |
| elevators | 2 | 18 | 16599 | 8.501 | 0.512 | 1.169 | 0.07 | 0.278 | 0.017 | 1.799 | 0.108 | 0.294 | 0.018 |
| riccardo | 2 | 4296 | 20000 | 5.994 | 0.3 | 1.369 | 0.111 | 0.325 | 0.018 | 2.21 | 0.111 | 0.358 | 0.018 |
| volkert | 10 | 180 | 58310 | 17.448 | 0.299 | 3.931 | 0.067 | 0.772 | 0.013 | 6.336 | 0.109 | 0.897 | 0.015 |
| higgs | 3 | 42 | 67557 | 21.991 | 0.326 | 6.449 | 0.095 | 1.356 | 0.02 | 10.404 | 0.154 | 1.517 | 0.022 |
| connect-4 | 2 | 28 | 98050 | 19.966 | 0.204 | 4.491 | 0.046 | 0.922 | 0.009 | 7.064 | 0.072 | 0.992 | 0.01 |
| BNG(vote) | 2 | 16 | 131072 | 28.229 | 0.215 | 8.898 | 0.068 | 1.636 | 0.012 | 14.25 | 0.109 | 1.95 | 0.015 |
| albert | 2 | 78 | 425240 | 77.546 | 0.182 | 27.988 | 0.066 | 5.748 | 0.014 | 47.855 | 0.113 | 6.378 | 0.015 |
| airlines | 2 | 7 | 539383 | 85.146 | 0.158 | 36.797 | 0.068 | 6.751 | 0.013 | 61.477 | 0.114 | 8.123 | 0.015 |
| agrawal1 | 2 | 9 | 1000000 | 193.083 | 0.193 | 70.665 | 0.071 | 12.129 | 0.012 | 112.533 | 0.113 | 14.944 | 0.015 |
| BNG(labor) | 2 | 16 | 1000000 | 190.81 | 0.191 | 69.374 | 0.069 | 12.094 | 0.012 | 108.211 | 0.108 | 14.666 | 0.015 |
| poker-hand | 10 | 10 | 1025009 | 198.151 | 0.193 | 71.605 | 0.07 | 13.097 | 0.013 | 112.214 | 0.109 | 15.199 | 0.015 |
| click-prediction-small | 2 | 11 | 1997410 | 384.192 | 0.192 | 133.212 | 0.067 | 24.964 | 0.012 | 227.546 | 0.114 | 30.982 | 0.016 |
| Average | | | 222079 | 44.525 | 1.109 | 15.212 | 0.124 | 2.839 | 0.087 | 24.814 | 0.162 | 3.402 | 0.09 |

Table 11: **TuneTables-medium and TuneTables-light are substantially faster with only a modest decrease in accuracy.** We compare the average accuracy and runtime in seconds of three versions of TuneTables, across all 19 datasets of size $< 50$K, as well as all 29 datasets. All datasets above the line were included in Table 1.

| Dataset | TuneTables-standard | | TuneTables-medium | | TuneTables-light | |
|---|---|---|---|---|---|---|
| | Acc. | Runtime | Acc. | Runtime | Acc. | Runtime |
| breast-cancer | 0.770 | 65 | 0.782 | 36 | 0.747 | 30 |
| heart-c | 0.903 | 66 | 0.892 | 27 | 0.892 | 23 |
| ecoli | 0.843 | 66 | 0.833 | 36 | 0.833 | 30 |
| colic | 0.892 | 66 | 0.892 | 27 | 0.892 | 23 |
| dresses-sales | 0.580 | 122 | 0.600 | 81 | 0.600 | 37 |
| climate | 0.951 | 97 | 0.963 | 38 | 0.963 | 30 |
| cylinder-bands | 0.846 | 82 | 0.870 | 41 | 0.870 | 34 |
| balance-scale | 0.995 | 55 | 0.995 | 26 | 0.995 | 22 |
| blood-transfusion | 0.782 | 56 | 0.773 | 31 | 0.760 | 30 |
| cmc | 0.556 | 109 | 0.536 | 32 | 0.514 | 44 |
| kc-1 | 0.856 | 187 | 0.843 | 68 | 0.842 | 50 |
| bioresponse | 0.798 | 3012 | 0.776 | 2374 | 0.766 | 653 |
| christine | 0.755 | 3920 | 0.740 | 3254 | 0.724 | 1366 |
| dilbert | 0.992 | 3749 | 0.990 | 2988 | 0.896 | 3107 |
| robert | 0.414 | 2397 | 0.420 | 1433 | 0.313 | 767 |
| har | 0.981 | 2657 | 0.985 | 2287 | 0.963 | 568 |
| eeg-eye-state | 0.986 | 1929 | 0.978 | 1752 | 0.938 | 213 |
| elevators | 0.900 | 1297 | 0.898 | 1110 | 0.897 | 211 |
| riccardo | 0.995 | 5247 | 0.995 | 3852 | 0.987 | 1309 |
| volkert | 0.693 | 6331 | 0.707 | 5872 | 0.562 | 1116 |
| connect-4 | 0.817 | 5395 | 0.813 | 4699 | 0.682 | 471 |
| higgs | 0.714 | 4084 | 0.714 | 3153 | 0.648 | 719 |
| BNG (vote) | 0.974 | 2493 | 0.973 | 517 | 0.970 | 113 |
| albert | 0.658 | 17518 | 0.654 | 2056 | 0.642 | 350 |
| airlines | 0.653 | 44434 | 0.643 | 2230 | 0.601 | 352 |
| BNG (labor) | 0.967 | 7717 | 0.963 | 3942 | 0.956 | 641 |
| agrawal1 | 0.950 | 45504 | 0.950 | 2131 | 0.949 | 531 |
| poker-hand | 1.000 | 10471 | 0.998 | 8184 | 0.575 | 630 |
| click-prediction-small | 0.837 | 33148 | 0.835 | 3050 | 0.834 | 635 |
| Average | 0.830 | 6975 | 0.828 | 1907 | 0.787 | 486 |

Table 12: **Ablation on TuneTables variants.** Comparison of the different variants of our method (TabPFN-PT) and the entire backbone fine-tuning (TabPFN-FT) to TuneTables itself on the LARGESCALETABLES benchmark. TuneTables is our full method described in Section 3. For each variant, we show its test accuracy. For all methods in this table except for TuneTables, there is no hyperparameter optimization (HPO). For TuneTables, we use our standard grid search.

| | TabPFN-FT | TabPFN-PT | TabPFN-PT | TabPFN-PT | TabPFN-PT | TabPFN-PT | TabPFN-PT | TuneTables |
|---|---|---|---|---|---|---|---|---|
| **Context in Training** | ✓ | | | ✓ | ✓ | ✓ | ✓ | Varied |
| **Context in Inference** | ✓ | ✓ | | ✓ | | ✓ | ✓ | Varied |
| **Tuned prompt length** | N/A | 1000 | 1000 | 1000 | 1000 | 100 | 10 | Varied |
| Agrawal1 | 0.946 | 0.948 | 0.950 | 0.950 | 0.946 | 0.950 | 0.949 | 0.950 |
| BNG(labor) | 0.931 | 0.961 | 0.965 | 0.965 | 0.966 | 0.965 | 0.960 | 0.967 |
| BNG(vote) | 0.964 | 0.971 | 0.974 | 0.973 | 0.973 | 0.974 | 0.971 | 0.974 |
| Bioresponse | 0.754 | 0.756 | 0.760 | 0.747 | 0.668 | 0.729 | 0.748 | 0.798 |
| Click_prediction_small | 0.834 | 0.834 | 0.834 | 0.835 | 0.835 | 0.835 | 0.833 | 0.837 |
| airlines | 0.588 | 0.628 | 0.645 | 0.643 | 0.639 | 0.640 | 0.628 | 0.653 |
| albert | 0.605 | 0.646 | 0.648 | 0.651 | 0.651 | 0.657 | 0.648 | 0.658 |
| balance-scale | 0.937 | 0.909 | 0.944 | 0.940 | 0.456 | 0.932 | 0.933 | 0.995 |
| blood-transfusion | 0.787 | 0.760 | 0.785 | 0.764 | 0.691 | 0.773 | 0.771 | 0.782 |
| breast-cancer | 0.730 | 0.743 | 0.713 | 0.741 | 0.661 | 0.724 | 0.718 | 0.770 |
| christine | 0.711 | 0.723 | 0.716 | 0.714 | 0.685 | 0.712 | 0.700 | 0.755 |
| climate-model | 0.947 | 0.938 | 0.938 | 0.941 | 0.666 | 0.923 | 0.938 | 0.951 |
| cmc | 0.542 | 0.525 | 0.559 | 0.540 | 0.436 | 0.532 | 0.548 | 0.556 |
| colic | 0.888 | 0.871 | 0.880 | 0.816 | 0.563 | 0.870 | 0.842 | 0.892 |
| connect-4 | 0.678 | 0.787 | 0.810 | 0.795 | 0.804 | 0.767 | 0.679 | 0.817 |
| cylinder-bands | 0.842 | 0.791 | 0.759 | 0.870 | 0.556 | 0.820 | 0.856 | 0.846 |
| dilbert | 0.825 | 0.934 | 0.941 | 0.895 | 0.847 | 0.869 | 0.834 | 0.992 |
| dresses-sales | 0.550 | 0.573 | 0.533 | 0.567 | 0.553 | 0.560 | 0.577 | 0.580 |
| ecoli | 0.814 | 0.765 | 0.775 | 0.823 | 0.314 | 0.833 | 0.823 | 0.843 |
| eeg-eye-state | 0.579 | 0.626 | 0.651 | 0.651 | 0.654 | 0.616 | 0.601 | 0.986 |
| elevators | 0.894 | 0.895 | 0.899 | 0.896 | 0.854 | 0.896 | 0.892 | 0.902 |
| har | 0.940 | 0.980 | 0.982 | 0.971 | 0.946 | 0.962 | 0.945 | 0.981 |
| heart-c | 0.863 | 0.839 | 0.860 | 0.806 | 0.718 | 0.838 | 0.855 | 0.903 |
| higgs | 0.638 | 0.700 | 0.703 | 0.696 | 0.700 | 0.698 | 0.675 | 0.714 |
| kc1 | 0.843 | 0.848 | 0.843 | 0.853 | 0.218 | 0.860 | 0.851 | 0.856 |
| poker-hand | 0.601 | 0.479 | 0.993 | 0.982 | 0.985 | 0.690 | 0.572 | 1.000 |
| riccardo | 0.918 | 0.990 | 0.993 | 0.990 | 0.990 | 0.960 | 0.936 | 0.995 |
| robert | 0.303 | 0.394 | 0.404 | 0.386 | 0.369 | 0.354 | 0.311 | 0.414 |
| volkert | 0.487 | 0.621 | 0.647 | 0.631 | 0.649 | 0.605 | 0.540 | 0.693 |
| Average | 0.757 | 0.774 | 0.797 | 0.794 | 0.689 | 0.777 | 0.763 | 0.830 |

Table 13: **Ensembled models outperform models trained on a single tuned prompt; with ensembling, the training and testing without real-data context (NC) setting matches or exceeds the setting with context (C).** Runs with prompt tuned only once are noted as TabPFN-PT, and ensembles of such runs as TabPFN-PT-Ens. Although the improvements are often quite small, we find that ensembles generally outperform single tuned prompts in both C and NC settings. We also find that ensembles trained without additional real data context at train time or test time are as good or better than ensembles trained and tested with real data context on datasets larger than 3000 samples.

| Dataset | TabPFN-PT-C | TabPFN-PT-Ens-C | TabPFN-PT-NC | TabPFN-PT-Ens-NC |
|---|---|---|---|---|
| agrawal1 | 0.949 | **0.95** | **0.95** | 0.949 |
| airlines | 0.645 | **0.649** | 0.645 | 0.646 |
| albert | 0.657 | **0.66** | 0.648 | **0.66** |
| balance-scale | 0.921 | 0.968 | **0.984** | 0.952 |
| bioresponse | 0.763 | **0.795** | 0.776 | 0.776 |
| blood-transfusion | 0.813 | **0.84** | 0.827 | 0.747 |
| BNG (labor) | 0.965 | 0.966 | 0.965 | **0.967** |
| BNG (vote) | 0.974 | 0.976 | 0.975 | **0.977** |
| breast-cancer | **0.793** | **0.793** | 0.759 | 0.69 |
| car | 0.96 | 0.971 | **0.977** | 0.965 |
| christine | **0.76** | 0.738 | 0.734 | 0.756 |
| click-prediction-small | 0.834 | **0.836** | 0.834 | **0.836** |
| climate | 0.926 | 0.944 | 0.963 | **0.981** |
| cmc | 0.534 | **0.581** | 0.547 | 0.446 |
| colic | 0.811 | **0.892** | 0.865 | 0.865 |
| connect-4 | 0.796 | **0.814** | 0.808 | 0.812 |
| cylinder-bands | 0.815 | 0.815 | **0.926** | 0.778 |
| dilbert | 0.87 | 0.951 | 0.948 | **0.968** |
| dresses-sales | 0.6 | **0.68** | 0.66 | 0.6 |
| eeg-eye-state | 0.74 | 0.977 | 0.666 | **0.983** |
| elevators | 0.903 | **0.908** | 0.902 | 0.902 |
| har | 0.94 | 0.984 | 0.976 | **0.987** |
| heart-c | 0.871 | **0.903** | 0.871 | 0.871 |
| higgs | 0.695 | **0.712** | 0.709 | 0.709 |
| kc-1 | 0.867 | **0.872** | 0.867 | **0.872** |
| poker-hand | **1** | **1** | 0.992 | **1** |
| riccardo | 0.991 | **0.996** | 0.994 | **0.996** |
| robert | 0.391 | 0.415 | 0.421 | **0.444** |
| volkert | 0.633 | 0.665 | 0.658 | **0.672** |
| Average | 0.807 | 0.836 | 0.822 | 0.821 |

Table 14: **TuneTables is competitive with strong baselines on a range of regression datasets.** In this table, we compare 7 algorithms on 15 regression datasets from [51]. The terminology and methods of this table follow Table 1, except that we report mean $R^2$ score instead of mean accuracy as our primary statistic. TuneTables ties with XGBoost for the number of wins, has the highest average $R^2$ score of any algorithm, and the second-highest mean rank after LightGBM.

| Model | Mean R2 | Mean Rank | Mean Z-Scores | Std Z-Scores | Med Z-Scores | Number of Wins |
|---|---|---|---|---|---|---|
| TuneTables | 0.603 | 3.533 | 0.792 | 0.448 | 0.768 | 4.0 |
| LightGBM | 0.536 | 3.333 | 0.640 | 0.472 | 0.660 | 1.0 |
| CatBoost | 0.508 | 3.800 | 0.431 | 0.933 | 0.576 | 2.0 |
| XGBoost | 0.386 | 4.000 | 0.116 | 1.009 | 0.405 | 3.0 |
| MLP | 0.359 | 5.267 | -0.009 | 0.673 | 0.109 | 1.0 |
| STG | 0.332 | 6.067 | -0.369 | 0.864 | -0.169 | 2.0 |
| TabNet | 0.239 | 5.867 | -0.107 | 0.747 | 0.099 | 0.0 |
| VIME | -0.007 | 7.733 | -1.072 | 1.054 | -1.159 | 0.0 |
| DeepFM | -7.954 | 5.400 | -0.422 | 1.248 | 0.027 | 2.0 |

Table 15: **TuneTables matches SOTA algorithms on small and medium-sized datasets.** In this table, we compare algorithms over 19 datasets in LARGESCALETABLES with at most 50 000 samples. For each algorithm, we compute its mean accuracy, mean runtime, and mean rank in terms of accuracy. We also compute the mean Z-score, computed by normalizing the set of results on each dataset (by mean 0 std. 1), so that each dataset has the same weight, and averaging each algorithm's normalized performances. Std. Z-Score is computed with respect to random splits and averaged across datasets. Fractional num. wins values are averaged over three splits per dataset, and reflect the presence of multi-way ties on certain splits. This table is similar to Table 1, but the benchmark is LARGESCALETABLES , and the search spaces for XGBoost and CatBoost are expanded to include more trees.

| Method | Mean Acc. | Mean Rank | Mean Z-Score | Std. Z-Score | Med. Z-Score | Num. Wins | Runtime |
|---|---|---|---|---|---|---|---|
| TuneTables | 0.831 | 2.263 | 0.471 | 0.536 | 0.614 | 9.833 | 1325 |
| CatBoost | 0.827 | 2.579 | 0.320 | 0.644 | 0.558 | 4.500 | 915 |
| XGBoost | 0.825 | 2.684 | 0.267 | 0.625 | 0.720 | 3.500 | 304 |
| rtdl_ResNet | 0.790 | 3.561 | -0.341 | 0.461 | -0.408 | 0.333 | 500 |
| rtdl_MLP | 0.755 | 3.912 | -0.717 | 0.678 | -0.897 | 0.833 | 374 |

Table 16: **Comparison of top-performing methods on the LARGESCALETABLES benchmark.** For each algorithm, we show its test accuracy and runtime (seconds). We also show the number of samples in each dataset. All datasets above the line were included in Table 15. This table is similar to Table 8, but the search spaces for XGBoost and CatBoost are expanded to include more trees.

| Dataset | Size | TabPFN | | TuneTables | | CatBoost | | XGBoost | |
|---|---|---|---|---|---|---|---|---|---|
| | | Acc. | Runtime | Acc. | Runtime | Acc. | Runtime | Acc. | Runtime |
| breast-cancer | 286 | 0.765 | 29 | 0.770 | 65 | 0.736 | 190 | 0.724 | 23 |
| heart-c | 303 | 0.848 | 40 | 0.903 | 66 | 0.839 | 164 | 0.849 | 27 |
| ecoli | 336 | 0.848 | 30 | 0.843 | 66 | 0.892 | 394 | 0.843 | 72 |
| colic | 368 | 0.856 | 39 | 0.892 | 66 | 0.874 | 458 | 0.883 | 27 |
| dresses-sales | 500 | 0.578 | 41 | 0.580 | 122 | 0.613 | 292 | 0.660 | 30 |
| cylinder-bands | 540 | 0.800 | 41 | 0.846 | 82 | 0.827 | 552 | 0.846 | 35 |
| climate | 540 | 0.959 | 59 | 0.951 | 97 | 0.951 | 282 | 0.938 | 29 |
| balance-scale | 625 | 0.990 | 29 | 0.995 | 55 | 0.931 | 241 | 0.894 | 112 |
| blood-transfusion | 748 | 0.801 | 25 | 0.782 | 56 | 0.756 | 92 | 0.760 | 27 |
| cmc | 1473 | 0.554 | 91 | 0.556 | 109 | 0.547 | 224 | 0.552 | 58 |
| kc-1 | 2109 | 0.862 | 168 | 0.856 | 187 | 0.855 | 138 | 0.853 | 40 |
| bioresponse | 3151 | 0.797 | 638 | 0.798 | 3012 | 0.791 | 618 | 0.797 | 131 |
| christine | 5418 | 0.742 | 666 | 0.755 | 3920 | 0.735 | 234 | 0.742 | 158 |
| robert | 10000 | 0.250 | 964 | 0.414 | 2397 | 0.534 | 2934 | 0.524 | 3294 |
| dilbert | 10000 | 0.922 | 761 | 0.992 | 3749 | 0.987 | 3185 | 0.986 | 779 |
| har | 10299 | 0.936 | 370 | 0.981 | 2657 | 0.993 | 3004 | 0.995 | 251 |
| eeg-eye-state | 14980 | 0.940 | 178 | 0.986 | 1929 | 0.957 | 2594 | 0.955 | 116 |
| elevators | 16599 | 0.902 | 186 | 0.902 | 1297 | 0.894 | 1316 | 0.896 | 103 |
| riccardo | 20000 | 0.922 | 1395 | 0.995 | 5247 | 0.996 | 473 | 0.996 | 468 |
| volkert | 58310 | 0.567 | 459 | 0.693 | 6331 | 0.715 | 2364 | 0.711 | 1550 |
| higgs | 67557 | 0.671 | 931 | 0.714 | 4084 | 0.729 | 222 | 0.726 | 128 |
| connect-4 | 98050 | 0.668 | 696 | 0.817 | 5395 | 0.864 | 3492 | 0.860 | 593 |
| BNG (vote) | 131072 | 0.968 | 1976 | 0.974 | 2493 | 0.975 | 42 | 0.975 | 76 |
| albert | 425240 | 0.642 | 2363 | 0.658 | 17518 | 0.709 | 4001 | 0.694 | 529 |
| airlines | 539383 | 0.600 | 2602 | 0.653 | 44434 | 0.671 | 2023 | 0.673 | 305 |
| BNG (labor) | 1000000 | 0.937 | 5518 | 0.967 | 7717 | 0.971 | 226 | 0.970 | 184 |
| agrawal1 | 1000000 | 0.948 | 5158 | 0.950 | 45504 | 0.951 | 142 | 0.951 | 98 |
| poker-hand | 1025009 | 0.531 | 2423 | 1.000 | 10471 | 1.000 | 6039 | 0.997 | 3469 |
| click-prediction-small | 1997410 | 0.833 | 10421 | 0.837 | 33148 | 0.842 | 84 | 0.843 | 102 |
| Average | | 0.781 | 1321 | 0.830 | 6975 | 0.832 | 1242 | 0.831 | 703 |

