# OpenReview forum: "TuneTables: Context Optimization for Scalable Prior-Data Fitted Networks"
_NeurIPS.cc/2024/Conference — NeurIPS 2024 poster_

### Official Review · Reviewer_cmgc · 2024-07-02

**Soundness:** 3
**Presentation:** 3
**Contribution:** 2
**Rating:** 6
**Confidence:** 4

**Summary:**

The paper proposes a method to compress large training sets into smaller contexts through soft prompt tuning for prior-data fitted networks. The proposed techniques relax the constraints of PFNs by allowing for an increased number of features, more in-context training examples, and a greater variety of classes. The authors show that the proposed methods achieve state-of-the-art performance across a wide range of datasets. In addition, the proposed method can also be used to optimize a fairness objective.

**Strengths:**

1. The paper is well-organized and easy to follow, with the problem well-motivated.
2. The paper tackles an important limitation of PFNs, which can inspire follow-up studies and enhance the practical usability of PFNs.
3. The authors conduct extensive experiments that study the advantages and limitations of the proposed method.

**Weaknesses:**

1. The proposed methods to deal with an increased number of features, training examples, and classes are disconnected, and some of the techniques are not new. The feature selection methods are classical methods that are widely used in machine learning and data sciences. Freezing the encoder of a network and fitting a new decoder is also very common in model finetuning.
2. The proposed method shows close performance to GBDTs, while it is much more computationally expensive.
3. The presentation of the results needs to be improved. In Figure 2, the lines are overlapping, making it hard to distinguish. Figure 3 is also confusing without a clear explanation of the y-axis.

**Questions:**

1. For tabular data, do you assume the type of the attributes (numerical, categorical, etc.)?
2. Do you have comparisons with previous works on data summarization?
3. For bias mitigation, I suggest also including a baseline of the base TabPFN while debiasing the selected training data.
4. Could you explain why PFNs only support a fixed number of classes?

**Limitations:**

yes

---

> ### Author Rebuttal · Authors · 2024-08-07
>
> Thank you for your detailed review. We appreciate that you found our paper well organized and easy to follow, and our problem well-motivated. We address each of your questions below:
>
> **W1: On novelty**
>
> Thank you for raising this point. We note that our novelty lies in applying prompt tuning to PFNs, and that we fill a major gap for PFNs: the issue of scaling beyond their various capacity limits. We respectfully note that the NeurIPS reviewer guidelines includes novel combinations of well-known techniques to be included in originality, and overall there has been a trend towards emphasizing impact rather than novelty.
>
> **W2: Computational expense**
>
> Thanks for bringing this up. We discuss this point as a limitation in Sec. 8 of our paper. However, we also note that, given the novelty of our method and the ongoing efforts to optimize the runtime of transformer-based architectures such as TuneTables, we can reasonably expect future research to further improve accuracy-runtime tradeoff of our method. Moreover, our method shows complementary strengths with GBDT, as each method outperforms on specific datasets.
>
> **W3: Presentation of results**
>
> We thank the reviewer for pointing out the improvements that can be made to Figure 2. We have created a wider version of Figure 2 such that the vertical lines for the algorithms no longer overlap, which we added to the paper. We also added the following to the caption of Figure 3:
>
> > The colorbar on the y axis represents the comparative change in per-dataset accuracy between two algorithms (A: blue, B: red). Positive numbers represent the absolute gain in accuracy of B w.r.t. A, negative numbers represent the absolute gain in accuracy of A w.r.t. B.
>
> **Q1: Assuming attributes**
>
> We do assume the basic type of each feature (numerical or categorical) on our tabular data.
>
> **Q2: Data summarization**
>
> We thank the reviewer for this question on data summarization. While preparing this rebuttal, we have completed additional experiments on several different methods for feature summarization using our own benchmarks (please see our global response part C), as this provides a realistic estimate of how feature summarization methods perform in this particular setting. Our results for sketching summarization are discussed in Appendix Table 5; there, we observe that no sketching method outperforms random search for most dataset-algorithm pairs. We hope this extension of results addresses your concern.
>
> **Q3: Bias mitigation**
>
> We thank the reviewer for making this suggestion. ​​In Sec. 5 of our paper, we cite fairness optimization as one particular example of a more general class of problems to which TuneTables is applicable, namely, multi-objective optimization problems. While in the particular case of fairness it might be possible to attain better results with TabPFN by debiasing the training data, this would not give us more insight into multi-objective optimization in TuneTables, as not all multi-objective optimization problems are amenable to such interventions. We nevertheless consider this an important and useful direction for future research into fairness with TuneTables.
>
> **Q4: Fixed number of classes**
>
> Thanks for this question. The TabPFN authors provide three reasons in their paper why they choose to fix the number of classes at 10; see below.
>
> “We focus on small datasets because (1) small datasets are often encountered in real-world applications (Dua and Graff, 2017), (2) existing DL methods are most limited in this domain (Grinsztajn et al., 2022) and (3) the TabPFN would be significantly more expensive to train
> and evaluate for larger dataset.”
>
> Thank you very much, once again, for your excellent comments. We respectfully ask that if you feel more positively about our paper, to please consider updating your score. If not, please let us know what can be further improved; we are happy to continue the discussion any time until the end of the discussion period. Thank you!

---

> > ### Comment · Reviewer_cmgc · 2024-08-11
> >
> > Thanks for the response. I will maintain my current score.

---

> > > ### Author Response · Authors · 2024-08-12
> > >
> > > Thanks very much for the discussion, and for considering our comments!

---

### Official Review · Reviewer_7Ntu · 2024-07-09

**Soundness:** 3
**Presentation:** 1
**Contribution:** 3
**Rating:** 6
**Confidence:** 4

**Summary:**

The paper proposes TuneTables, a method for improving the performance of PFNs on large datasets with a variable number of features and classes. TuneTables uses prompt tuning (fine-tuning) to learn a small set of parameters (context / synthetic datapoints). Depending on the qualities of the dataset, a feature subselection method or a new decoder may be used. The experiments show that TuneTables outperforms TabPFN on larger datasets. In the analysis, the paper shows that bias (specifically demographic parity) can be mitigated with a regularizer.

**Strengths:**

- Impressive number of experiments that show TuneTables outperforms TabPFN
 - Soft Prompt Tuning for TabPFN is an interesting idea

**Weaknesses:**

My main concern with this paper is its presentation.

Most notably, I found it hard to read Section 4. It is particularly dense. There are a lot of Table/Figure references (in total 8), all of which are not on the same page, so a reader would need to flip back and forth between pages to find the table and figures.
Furthermore, many of the references in Section 4 point to various resources in the Appendix.
For example, many analyses are mentioned without the tables to back them up in the main paper. Instead, tables in the appendices are linked. The main paper should be complete. The reader should be able to draw conclusions from the provided experiments (tables and figures). In this case, the conclusions are mentioned but the Appendix is used as additional space for the figures and tables.

Here are just some of the examples:
 - "(see Appendix Table 7 for the results). [...] we find that TuneTables achieves [...]"
 - "In Appendix Table 6 we show that [...]"
 - "At inference time, TuneTables is significantly faster than TabPFN; see Table 9" (Table 9 is in the Appendix).
 - "ablate the tuned prompt size and the sue of ensembles in Table 11 and Table 12, finding that smaller" (Tables 11 and 12 are in the Appendix).

Interestingly, Section 5 and 6 are not dense and very easy to read which further makes the paper awkward to read (going from a dense Section 4 to a super not dense Section 5 and 6). One way to tackle this is to shorten Sections 5 and 6, making more space for Section 4, allowing for a small subset of Tables in the Appendix to be moved to the main paper which would make Section 4 easier to read.

Some additional notes regarding presentation:
 - "TuneTabes-hard" is the name of a dataset. The dataset name is too close to that of the method: "TuneTables". The names should ideally be easily distinguishable
 - (Figure 3) "subset of [48]" -- Is there a dataset name for [48]? It would be easier to read if so. "[48]" is used to reference the paper but is also used as the "name" of the dataset.
 - (Figure 3) "reduces or reverses TabPFN's limitations" -> "addresses TabPFN's limitations"
 - "One limitation of [48] Table 1" -- I'm guessing this is referring to Table 1 from [48], but this is confusing as there is also a Table 1 in this paper.
 - "GBDTs" - this term is mentioned several times without clearly defining what it is an acronym for, i.e., Gradient Boosted Decision Trees.
 - "Motivated by the limitations of sketching for large contexts" -- To the best of my knowledge, no limitations of sketching were mentioned before this point, so it is unclear what it is being referred to
 - Section 7: "PFNs are neural processes (NPs) [52]" -- If I understand correctly, it appears that NPs and PFNs are being explored separately but are similar models if not the same. As mentioned in the paper, there have been several neural processes works that tackle similar issues with scaling transformer-based models, so it would be nice if the paper connected NPs and PFNs more. For example, clarify that PFNs are neural processes earlier. The focus of the paper on PFNs and the brief mention of "(similar to works on neural processes; see Section 7)" suggested to me that PFNs and NPs are considerably different for the majority of the paper until I read Section 7.

Minor:
 - Typo: "subset of [48]on" -> "subset of [48] on" (missing a space)
 - Section 4 is titled "Experiments". Section 5 and Section 6 are experiment/analyses-focused, so it's more accurate for them to be subsections of Section 4.

**Questions:**

See Weaknesses.


"[TuneTables] also does not improve on TabPFN for small datasets (fewer than 1000 samples, 100 features and 10 classes); we postulate this is a result of overfitting." -- Could you clarify why you postulate it is due to overfitting? On page 4, it is mentioned that "if a dataset is small enough to run with the original zero-shot version of TabPFN". Wouldn't you just use the original zero-shot version of TabPFN in this case (without any tuning)?

"(b) if there are too many features" -- Could you clarify how you measured what constitutes "too many"?

"limited to around 3000 by conventional GPU sizes." -- Is this assuming a batch size of "1" with "3000" context tokens? Or a larger batch size?


"memory requirements scale quadratically with the context length" -- This is not necessarily the case. At inference time, attention can be computed recurrently (Rabe et al., 2021; Feng et al., 2024), resulting in transformers only needing linear memory with the context length (albeit the amount of computation in total is still quadratic).

---

"Self-attention Does Not Need $O(n^2)$ Memory", MN. Rabe, C. Staats, arXiv:2112.05682, 2021

"Attention as an RNN", L. Feng, F. Tung, H. Hajimirsadeghi, MO. Ahmed, Y. Bengio, G. Mori, arXiv:2405.13956, 2024

**Limitations:**

Yes

---

> ### Author Rebuttal · Authors · 2024-08-07
>
> Thank you for your detailed review. We appreciate that you found our idea interesting, and that you appreciated the large number of experiments we provide. We address each of your questions below:
>
> **W1: Presentation**
>
> We thank the reviewer for a comprehensive set of suggestions on how to improve the presentation of our work. We especially agree with you that the main paper should be complete by itself. We have thought through and made all of the changes to make this true and to address all of your comments. Because NeurIPS does not allow us to upload a modified version of our paper as part of the discussion, we cannot provide you with a highlighted manuscript with our changes; instead, we summarize the major points below.
>
> - We significantly compressed Sections 5 and 6 by having just one section titled “TuneTables extensions”, with mitigating bias and dataset understanding as two subsections, and substantially cutting down the text to have only the main points.
> - Even with the compression of Sections 5 and 6, we cannot add all of Tables 6-12 to the paper. In order to reduce the need for the reader to flip between pages while reading, we added one-sentence summaries of the contents of the main appendix tables referenced (6,7,9), where they are referenced in the Sec. 4. Here is an example of such a change:
>
> >In Appendix Table 6 we show that despite the large divergence from the PFN’s pretraining, TuneTables achieves a high rank, second only to XGBoost. -> In Appendix Table 6 we show that despite the large divergence from the PFN’s pretraining, TuneTables achieves a mean rank of 2.52, ahead of CatBoost and a ResNet, second only to XGBoost, whose mean rank is 2.0.
>
> - We move much of the ablation discussions in Sec 4 into the appendix next to tables 11 and 12, giving a high-level summary in Sec. 4
> - We now name the benchmark in our Table 1 as the “tabzilla benchmark suite” to be consistent with the language of recent work.
> - We also changed "reduces or reverses TabPFN's limitations" -> "addresses TabPFN's limitations", per the suggestion
> - Thanks for the suggestion; we renamed the “TuneTables-Hard” dataset to LargeScaleTables, to reduce confusion between the algorithm and the dataset
> - We have also made the smaller edits you suggest, including directly referencing neural processes earlier in the paper – we now write, “A recent breakthrough, prior-data fitted networks (PFNs) [33, 52] are a specific type of neural process (Garnelo et al., 2018) which learn to perform approximate Bayesian inference in a single forward pass using in-context learning.”
>
> **Q1: Overfitting**
>
> Thanks for the question. In our experiments, we found that TuneTables would sometimes attain higher 3-fold cross-validation accuracy compared to zero-shot TabPFN, but lower test accuracy, particularly when val and test sets were small. TabPFN, which was pretrained on large quantities of synthetic data and did not optimize using the val set, exhibited low val/test divergence on all datasets. We will add this comment to the final version of our paper.
>
> **Q2: Zero-shot TabPFN**
>
> Thanks for the question. Zero-shot TabPFN is indeed used as part of the TuneTables grid search on such datasets. The reason why we do not only use TabPFN is that the exact size at which TuneTables outperforms TabPFN is dataset-dependent (with an average transition of around 800 samples). We will add this comment to the final version of our paper.
>
> **Q3: Too many features**
>
> Thanks for the question. By ‘too many features’, we mean more than the 100 features which TabPFN can utilize without feature selection methods. We will clarify this in our final draft as well.
>
> **Q4: Batch size 1**
>
> Yes! The statement is assuming a batch size of 1. We conduct the relevant experiments on NVIDIA RTX8000 GPUs with 48GB of VRAM per GPU. We will clarify this in our final draft.
>
> **Q5: Memory requirements**
>
> We thank the reviewer for raising the topic of FlashAttention. Indeed, FlashAttention is memory-linear in sequence length. However, please note that it has not yet been implemented in the public release of TabPFN -- it may be integrated into a future release, and any benefits will carry over to our method. Please also note that while Flash Attention is extremely useful, it still has other limitations: inference time continues to be quadratic, and there are other overheads on GPU memory. Overall, there remains a considerable need for new algorithmic methods which can be scaled up to the sequence lengths required by large tabular datasets. We will add a clarification in our paper regarding this point.
>
> Thank you very much, once again, for your excellent comments. We respectfully ask that if you feel more positively about our paper, to please consider updating your score. If not, please let us know what can be further improved; we are happy to continue the discussion any time until the end of the discussion period. Thank you!

---

> ### Comment · Reviewer_7Ntu · 2024-08-07
>
> Thank you for the rebuttal. Although I am unable to verify the presentation improvements, I believe the authors' commitments and stated changes. I lean towards accepting the paper and have updated my score (4 -> 6) to reflect this.

---

> > ### Author Response · Authors · 2024-08-12
> >
> > Thank you for acknowledging our rebuttal and for taking the time to review our work!

---

### Official Review · Reviewer_FY4B · 2024-07-11

**Soundness:** 3
**Presentation:** 3
**Contribution:** 4
**Rating:** 7
**Confidence:** 4

**Summary:**

This paper aims at removing 3 issues with the recently introduced TabPFN model (a transformer pretrained on synthetic datasets, showing great in-context performance on actual small tabular datasets), namely that it can only take as input few samples, few features, and few classes. The authors manage to solve these issues by a) for classes, training a new decoder with the rest of TabPFN frozen b) for features, searching over a few simple feature reduction methods (mutal info, PCA..) c) for samples, the authors use prompt tuning: some embeddings are trained to improve TabPFN performance while going through a dataset which can be bigger than TabPFN acceptable context length. Combining all these improvement creates a new method named TuneTables.

Experiments on both a standard tabular benchmark and a new benchmark shows that TuneTables beats all their strong baselines for tabular data prediction, even on datasets with many samples, features or classes. While TuneTables is slower than Gradient Boosting Tree algorithms, the authors provides variants with different time-performance tradeoff, and suggest ways to speed up the model's inference time.

Finally, the authors shows that TuneTables can also be used to do things which were not possible to do with TabPFN, for instance optimizing a different objective for fairness, or having access to "summary points" for a specific dataset.

**Strengths:**

The contributions of the paper are important and novel: this can allow TabPFN to be used in a much bigger set of contexts.

The evaluation of the new TuneTables method is well-done: the method is compared to strong baselines (CatBoost, XGBoost, recent deep learning models, but also interesting variants like finetuning TabPFN) on a standard benchmark, with several aggregation metrics. Furthermore, the results of the method are impressive.

The paper is quite rich, with a lot of results and ablations.

The codebase seems quite easy to install and use (though I haven't actually run anything).

I like that the paper starts off by considering simple solutions in Section 2.1 (some of which they keep) before moving to more complex solutions.

The paper is well written.

The authors are forthcoming when talking about the weaknesses of their method, for instance the slower runtime of their method compared to GBDTs.

**Weaknesses:**

Important details are missing, for instance the hyperparameter spaces used for the baselines and for TuneTables. This is an important part of the evaluation (which is not easy to find in the codebase), and I'm interested in how the space depends on the dataset.

Some information is hard to find. I would recommend pointing directly to the relevant results (instead of a broad section like section 4). I would also recommend a table of content for the appendix. For instance I was quite interested in understanding "In Section 4, we find that all sketching and feature selection methods plateau in performance well before approaching parity with GBDTs, which motivates our investigation of new scaling techniques.". This lead me to follow: section 2.1 --> section 4 --> find paragraph on sketching --> appendix C1 --> table 5 --> not sure how to interpret.

Relatedly, all ablations should be reported using aggregated scores, in addition to individual results. This would also make it easier to go through the many (which is good!) ablations.

I also wonder why most ablations and variants are done on the new benchmark TuneTables-hard instead of the more standard TabZilla benchmark. I think it makes sense in some cases, but for instance for TuneTables-medium and light I would be interested in seeing the performance on the TabZilla benchmark.

For these two variants, I would also be interested in seeing a complete performance and runtime comparison with baselines like GBDTs in addition to what you say in the main text. This would help the reader to better understand the time-performance pareto frontier.

**Questions:**

> While PFN accuracy can improve with more real-data samples at inference time [33, 55], the memory requirements scale quadratically with the context length

Isn't it linear with FlashAttention (which is used by TabPFN if I remember correctly).

Which hyperparameter spaces are you using for each model?

You say several times that you do a grid search on TuneTables hyperparameters, using 30 steps. But it seems that the hyperparameters space is bigger than 30 possibilities. Are you doing a random search?

How were the datasets in tunestable-hard chosen? (also I wonder whether the name may be a bit misleading because it first got the impression that it contained only dataset with a lot of samples or features)

In Table 11, are the results compared for a given number of HPO steps?

I'm a bit confused about how ensembling is working for TuneTables, with some sentences giving me the impression that you're taking a tuned prompt and permuting the order / labels (à la TabPFN), and other sentences giving me the impression that you're actually training several tuned prompts ("a step constituting either a new tuned prompt fitted from scratch
686 or a new member of a tuned prompt ensemble")

how are coresets computed?

**Limitations:**

The authors have adequately addressed limitations of their work.

---

> ### Author Rebuttal · Authors · 2024-08-07
>
> Thank you for your detailed review. We appreciate that you found our paper important and novel! We address each of your questions below:
>
> **W1: Hyperparameter spaces**
>
> We thank the reviewer for raising this point. We add TuneTables hyperparameters in the PDF attached to the global comment. For all baselines, we use the same hyperparameter ranges as the Tabzilla paper with one exception: we increased the range of tree depth by a factor of 100, since a high depth is much more appropriate for very large datasets. The tabzilla authors point the reader to [this file](https://github.com/naszilla/tabzilla/tree/main/TabZilla/models) in order to see hyperparameter ranges. We are happy to add all of this information directly into our paper.
>
> **W2: Arrangement of the paper**
>
> We thank you for drawing our attention to the indirection in Sec. 4. We will update this so that it refers to Table 5 directly. Regarding how to interpret Table 5, we clarify below:
>
> The columns labeled SKT / FTS / SMP list the best performing method for sketching, feature subsampling and label-aware sketching technique, respectively. Label-aware sketching refers to a strategy where we either sample instances proportionate to their labels, or we oversample minority classes with replacement to create a class-balanced distribution. While the choice of label-aware sketching strategy is often impactful (and we use it in TuneTables), and the choice of feature subselection method can be important for some datasets, in all but one case, no sketching method we test outperforms random sampling.
>
> The comment that “all sketching and feature selection methods plateau in performance well before approaching parity with GBDTs” is referring to the fact that in Table 5, TabPFN fails to match CatBoost’s performance on seven datasets, all of which are very large.
>
> In the final version of our paper, also as you suggest, we will incorporate a table of contents for our appendix, and we will add the above discussion near Table 5.
>
> **W3: Ablation aggregate scores**
>
> Thanks for this note! We will add relevant aggregate scores to all results tables in the main paper.
>
> **W4: TT-medium and TT-light**
>
> We thank you for raising this point about reporting TuneTables-medium and TuneTables-light results for all of TabZilla, including runtimes, in order to better understand the time-performance pareto frontier of our method. While preparing this rebuttal, we have completed these experiments on the 98 datasets from the tabzilla benchmark suite; (please see our global response; section A). We hope this extension of results helps to address your concern.
>
> **Q1: Memory requirements**
>
> We thank the reviewer for raising the topic of FlashAttention. For our reply, please refer to our comment on review 7Ntu, Q5.
>
> **Q2: Grid search**
>
> The space over which we conduct grid search in TuneTables is conditioned on metadata; # features, # samples, # classes. The feature and class splits are set by the limits of the particular TabPFN checkpoint we optimize. When we reach a leaf node, TuneTables-standard and TuneTables-medium conduct a grid search over a fixed range of configurations. The size of our search space is always < 30, and usually < 10. For feature-large datasets, TuneTables-light conducts additional optimization in the feature space prior to grid search. See the PDF attached to our global response for further details. For the final version of our paper, we will add this clarifying material to the appendix.
>
> **Q3: TuneTables-Hard criteria**
>
> We curated datasets with far more samples and features than in the standard datasets from tabzilla (max 45,000 samples) from OpenML to better understand the scaling challenges of PFNs, omitting image classification datasets. Finally, because we intended to include TabPFN in our analysis and every dataset we had curated was beyond the recommended limits of that model, we added several small datasets from tabzilla; since TabPFN generally outperformed boosted trees on these smaller datasets, and TuneTables extended TabPFN, we heuristically selected smaller datasets so as not to favor either TabPFN or boosted trees. For the final version of our paper, we will add these details to the appendix.
>
> **Q4: Table 11**
>
> We thank the reviewer for this question. In Table 11, we compare different variants of our method and entire backbone fine-tuning (TabPFN-FT) to TuneTables itself. For all methods in this table except for TuneTables, there is no hyperparameter optimization (HPO). For TuneTables, we use our standard grid search. For the final version of our paper, we will clarify this.
>
> **Q5: Ensembling**
>
> We thank the reviewer for this clarifying question about ensembling in TuneTables. In a TuneTables ensemble, each ensemble member fits its own tuned prompt to the data. Variance in the ensemble members is introduced by differences in the random initialization of the tuned prompt, as well as permuting the order of features and labels, a la TabPFN, one time before each tuned prompt is fitted. For the final version of our paper, we will clarify the sentence you cited in your analysis.
>
> **Q6: Coresets**
>
> For efficient coreset selection, we use a variant of [Farthest Point Sampling](https://ieeexplore.ieee.org/document/577129); after selecting an initial set of n=5 random points, we compute the distance of each point in the dataset to the set of already selected points. Next, we add to the set of selected points the point whose distance to the selected points is maximal. Finally, we update the distances of all the points according to the updated selected set; and continue iteratively. We will add these details to the final version.
>
> Thank you again for your excellent comments. We respectfully ask that if you now feel even more positively about our paper, to consider slightly increasing your score. We are happy to continue the discussion any time until the end of the discussion period. Thank you!

---

> > ### Comment · Reviewer_FY4B · 2024-08-11
> >
> > Thank you for your response, which clearly answers my questions!
> >
> > **TT-medium and TT-light**: thank you for providing the score of these models on the TabZilla benchmark. The results are interesting though I find the average accuracy to be hard to interpret, often leading to very close score (here XGBoost, TuneTable-Medium and TuneTable-Light). For the runtime, I'm also worried that it roughly follows the runtime for the biggest dataset, which is interesting but doest give the full picture. For the revised manuscript I would suggest metrics like "Runtime per 1000 samples", or reporting runtimes in different metadata feature bins.
> >
> > **Flash Attention**: the TabPFN code uses Pytorch's MultiHeadAttention, which should default to FlashAttention when possible.
> >
> > I'm keeping my Accept score.

---

> > > ### Author Response · Authors · 2024-08-12
> > >
> > > TT-medium and TT-light: Thank you for these further suggestions. For all methods in our table, we will report the runtime per 1000 samples for the TabZilla benchmark in the revised manuscript, binning the runtimes into four groups and then averaging within each bin: (FEATURE-LARGE, SAMPLE-LARGE), (FEATURE-LARGE, SAMPLE-SMALL), (FEATURE-SMALL, SAMPLE-LARGE), (FEATURE-SMALL, SAMPLE-SMALL).
> > >
> > > Flash Attention: we thank the reviewer for pointing this out. We will add an appropriate clarification in the revised manuscript.
> > >
> > > We thank the reviewer for their comments and a fruitful discussion.

---

### Official Review · Reviewer_YTTX · 2024-07-13

**Soundness:** 3
**Presentation:** 3
**Contribution:** 2
**Rating:** 5
**Confidence:** 5

**Summary:**

The paper introduces a novel parameter-efficient fine-tuning strategy for prior-data fitted networks (PFNs) called TuneTables. PFNs, similar to large language models, utilize pretraining and in-context learning to achieve strong performance on new tasks in a single forward pass. However, existing PFNs like TabPFN are limited to small datasets (less than 1000 samples). TuneTables addresses these limitations by compressing large datasets into a smaller learned context, thus significantly improving the performance of PFNs. Besides, this paper gave an demonstration of TuneTables as an interpretability tool and for mitigating biases by optimizing a fairness objective.

**Strengths:**

- TuneTables enhances the performance of PFNs on large datasets, surpassing other state-of-the-art models like CatBoost.

- Moreover, this paper provides comprehensive experiments and detailed analyses.

- The presentation is good, and the methodology is sound.

**Weaknesses:**

- Since the dataset was collected by the authors themselves, there are some aspects I am unsure about. In previous benchmarks, FT-Transformer generally outperforms SAINT, but on the authors' benchmark, it does not. Given that tabular data is diverse, it is possible to collect datasets that are more friendly to sample-interaction-based architectures like SAINT, PFN, and TableTunes.

- Excelformer, a well-known method that performs well on both small-scale and large-scale datasets, was not compared. I also suggest that the authors provide tests on Excelformer's large-scale datasets.

- The authors mentioned, "For all algorithms other than TuneTables, we perform light hyperparameter tuning by running one default setting and 29 iterations of random search using Optuna." This is clearly inconsistent with the original settings of most methods. I have concerns about the results.

- This method improves upon TabPFN. However, TabPFN can only be applied to classification tasks, neglecting the regression tasks. Additionally, as stated in Lines 82-88, there are issues like a fixed number of features and a fixed number of classes. These problems persist, even though the authors employed techniques like sketching and feature selection (not to mention that sketching and feature selection can also be applied to other methods, as done in Kaggle competitions, potentially improving their performances). All these factors undermine the significance and value of this paper.

**Questions:**

see above

**Limitations:**

The authors clearly summarized their limitations.

---

> ### Author Rebuttal · Authors · 2024-08-07
>
> Thank you for your detailed review. We appreciate that you found our paper to be presented well, and our methodology sound. We address each of your questions below:
>
> **W1: Chosen benchmarks**
>
> We thank the reviewer for raising this clarifying point about our benchmarks. Our main results in Tab. 1 are reported on a standard benchmark from the literature, [tabzilla](https://arxiv.org/abs/2305.02997). In the tabzilla benchmark, SAINT outperforms FT-Transformer; this is also the case in the largest-scale comparison in the well-known [Grinsztajn](https://arxiv.org/pdf/2207.08815) benchmark (see Fig. 1), where SAINT outperforms FT-Transformer outright on regression tasks and on classification tasks with few iterations of random search (there is no statistically significant difference after up to 100 iterations of random search on classification tasks). The benchmark introduced in our paper, TuneTables-Hard, was curated in part to illustrate TabPFN’s scaling limitations – for further discussion on this point, please see our response to FY4B, Q3.
>
> **W2: ExcelFormer**
>
> We thank you for raising this point about ExcelFormer, a recently published method. We have implemented this method for our rebuttal, and completed experiments on around half of the 98 datasets from the tabzilla benchmark suite; the initial results are intriguing, and we hope to extend them after the rebuttal period. Please see our global response -- tables B1 and B2; we hope this extension of results addresses your concern.
>
> **W3: Hyperparameter tuning**
>
> Experimental setup for hyperparameter tuning. This is a good question; thanks for allowing us to clarify! All of the algorithms come with their default set of hyperparameters used in the official implementation, and we used all of these settings. It is common in tabular papers to further improve performance for each dataset by conducting hyperparameter tuning (via k-fold cross validation) on a per-dataset basis. To be consistent with prior work such as Tabzilla, TabPFN, and [Grinsztajn et al](https://arxiv.org/pdf/2207.08815), we chose the best configuration among the default parameter sets and the additional 29 sets. We will clarify this in the paper, and we are happy to answer any further follow-up questions about our experimental setting.
>
> **W4: Regression tasks**
>
> We thank you for raising this point about regression tasks. TabPFN does provide a [server](https://github.com/automl/tabpfn-client) to run regression tasks, but to the best of our knowledge, no regression results have yet been made public by the authors. We have completed extensive preliminary regression experiments on TuneTables for this rebuttal; please see our global response, Table D. We hope that this significant extension of results helps to address your concern.
>
> Thank you very much, once again, for your excellent comments. We respectfully ask that if you feel more positively about our paper, to please consider updating your score. If not, please let us know what can be further improved; we are happy to continue the discussion any time until the end of the discussion period. Thank you!

---

> > ### Comment · Reviewer_YTTX · 2024-08-11
> >
> > Thank you for your response. Your answer has partially resolved my issue. I have increased the score.

---

> > > ### Author Response · Authors · 2024-08-12
> > >
> > > Thank you for acknowledging our rebuttal and for taking the time to review our work!

---

### Author Rebuttal · Authors · 2024-08-07

We thank all of the reviewers for their valuable feedback. Our work introduces TuneTables, allowing PFNs to scale by orders of magnitude and achieve strong performance on large datasets. We appreciate that the reviewers find our techniques important, interesting and novel (FY4B, 7Ntu, cmgc), with the empirical results of TuneTables (YTTX, FY4B, 7Ntu, cmgc), and comprehensible, easily understandable writing (cmgc, FY4B) listed as strengths. Following your suggestions, we highlight further improvements:
- (A) We compare TuneTables-medium and light to our baselines on the 98 datasets curated in the [TabZilla Benchmark Suite](https://arxiv.org/abs/2305.02997). TuneTables outperforms all algorithms at less than 60% of CatBoost’s runtime; TuneTables-light performs nearly identically to CatBoost at less than 20% of its runtime (although it is still slower than a highly optimized XGBoost)
- (B) We present new results for the ExcelFormer algorithm on the TabZilla Benchmark Suite
- (C) We present a new ablation on feature selection
- (D) We add regression experiments with TuneTables, utilizing 10 datasets (also sourced from TabZilla)

We would be very happy to keep the discussion going, addressing any points that remain unclear, or any new suggestions. Thanks again for your suggestions!

**(A) TUNETABLES VS CATBOOST AND XGBOOST ON THE TABZILLA BENCHMARK SUITE**

These results are reported for the main 98 datasets in the TabZilla Benchmark Suite, with all algorithms reporting results for all datasets. We report the aggregate (mean) values, averaged over three splits. Accuracy is on the test split. TuneTables outperforms all algorithms, while requiring less than 60% of CatBoost’s runtime.

| Algorithm     	| Accuracy | Runtime |
|------------------|----------------|----------|
| TuneTables   	| 0.861      	| 573  |
| CatBoost     	| 0.857      	| 1061 |
| TuneTables-medium| 0.855      	| 305  |
| XGBoost      	| 0.855      	| 57   |
| TuneTables-light | 0.854      	| 196  |

**(B) EXCELFORMER RESULTS**

Per Reviewer YTTX’s request, we have run ExcelFormer on the TabZilla Benchmark Suite; given the limited time we had for the rebuttal, we report here on a 50 dataset subset of the TabZilla Benchmark Suite in which ExcelFormer, CatBoost, TuneTables and XGBoost all ran successfully (B1), as well as a 21 dataset subset in which all algorithms ran successfully (B2). ExcelFormer generally performs well, and is among the strongest deep learning algorithms in this limited subset. For the final version of this paper, we will extend ExcelFormer to more datasets for a comprehensive picture.

**TABLE B1**

| alg_name	| Accuracy | Runtime |
|-------------|----------------|----------|
| TuneTables  | 0.867      	| 681  |
| CatBoost	| 0.864      	| 972  |
| XGBoost 	| 0.862      	| 68   |
| ExcelFormer | 0.852      	| 811  |

**TABLE B2**

| alg_name        	| Accuracy | Runtime |
|---------------------|----------------|----------|
| TuneTables   | 0.828      	| 13   |
| TabPFN.         	| 0.828      	| 69   |
| CatBoost        	| 0.819      	| 755  |
| NODE            	| 0.812      	| 2080 |
| XGBoost         	| 0.808      	| 47   |
| rtdl_ResNet     	| 0.808      	| 183  |
| SVM             	| 0.803      	| 1192 |
| ExcelFormer     	| 0.802      	| 371 |
| LightGBM        	| 0.801      	| 13  |
| RandomForest    	| 0.799      	| 9	|
| DANet           	| 0.798      	| 843 |
| SAINT           	| 0.797      	| 906  |
| rtdl_FTTransformer  | 0.795      	| 246 |
| TabNet          	| 0.786      	| 498  |
| DecisionTree    	| 0.774      	| 1	|
| rtdl_MLP        	| 0.770       	| 143  |
| LinearModel     	| 0.764      	| 1	|
| MLP             	| 0.749      	| 240  |
| STG             	| 0.752      	| 273  |
| KNN             	| 0.751      	| 3	|
| VIME            	| 0.726      	| 301  |

**(C) FEATURE SELECTION ABLATION**

| Model     | Metric | Random | Mutual Information | PCA  | PCA + Whitening | ICA  | Sparse Random Projection |
|-----------|--------|--------|--------------------|------|----------------|------|--------------------------|
| **TabPFN**| **Avg. Test Acc.**| 0.347  | 0.539              | 0.557| 0.557     | 0.565| 0.566                    |
| **CatBoost**| **Avg. Test Acc.**| 0.420  | 0.685              | 0.727| 0.727     | 0.726| 0.702                    |


We provide a preliminary ablation on our choice of feature selection methods in the paper, PCA and mutual information using three datasets from TabZilla. Similar to our findings in Appendix Table 5, we note that random feature selection is not a very strong baseline, particularly for TabPFN. We observe that PCA and mutual information are reasonable options, but in certain cases, other methods will perform better; in the interest of reducing our training time, we keep our search space small. We choose PCA and mutual information for TuneTables as they exhibit complementary strengths.

**(D) TUNETABLES REGRESSION RESULTS**

We report preliminary TuneTables regression results on 10 datasets from TabZilla. For these experiments, we train a new PFN from scratch on a new space of synthetic datasets designed for regression analysis, as we find this checkpoint performs better than TabPFN on average. We will add details of the model training to our final paper. We report non-normalized R2 scores on the test set, averaged over three shifts, as well as the average end-to-end runtime over three shifts. Unfortunately, the tabzilla authors have yet to release the raw results for their regression experiments, and we therefore were not able to compare these scores to any baseline models in time for the rebuttal – however, we will do so in the final draft of our paper.

In general, TuneTables performs well, with an average R2 score of .808 across all datasets. See the attached PDF for the full results.

---

### Decision · Program_Chairs · 2024-09-25

**Decision:**

Accept (poster)

**Comment:**

This paper represents a solid contribution to the field of tabular data analysis and thus deserves to be accepted to the main proceedings of NeurIPS. Reviewers in particular described the experimental results as extensive and mentioned the motivation was clear. There were some major concerns about writing raised by a reviewer that the authors have claimed to improved; I would like to stress that improvements in writing along those lines should be considered as a requirement for acceptance. The authors also provided another baseline at the request of a different reviewer and promised to flesh out the results fully for the main paper; I would expect this promise to be fulfilled as well.